# Dual-plasmonic Au@Cu$_7$S$_4$ yolk@shell nanocrystals for photocatalytic hydrogen production across visible to near infrared spectral region

Chun-Wen Tsao[1], Sudhakar Narra [2], Jui-Cheng Kao[1], Yu-Chang Lin [3],
Chun-Yi Chen[4], Yu-Cheng Chin[5], Ze-Jiung Huang[5], Wei-Hong Huang[6],
Chih-Chia Huang [5], Chih-Wei Luo [3,6,7], Jyh-Pin Chou [8], Shigenobu Ogata [9],
Masato Sone[4], Michael H. Huang [10], Tso-Fu Mark Chang [4] ✉,
Yu-Chieh Lo [1] ✉, Yan-Gu Lin [3] ✉, Eric Wei-Guang Diau [2,11] ✉ &
Yung-Jung Hsu [1,11,12] ✉

Near infrared energy remains untapped toward the maneuvering of entire solar spectrum harvesting for fulfilling the nuts and bolts of solar hydrogen production. We report the use of Au@Cu$_7$S$_4$ yolk@shell nanocrystals as dual-plasmonic photocatalysts to achieve remarkable hydrogen production under visible and near infrared illumination. Ultrafast spectroscopic data reveal the prevalence of long-lived charge separation states for Au@Cu$_7$S$_4$ under both visible and near infrared excitation. Combined with the advantageous features of yolk@shell nanostructures, Au@Cu$_7$S$_4$ achieves a peak quantum yield of 9.4% at 500 nm and a record-breaking quantum yield of 7.3% at 2200 nm for hydrogen production in the absence of additional co-catalysts. The design of a sustainable visible- and near infrared-responsive photocatalytic system is expected to inspire further widespread applications in solar fuel generation. In this work, the feasibility of exploiting the localized surface plasmon resonance property of self-doped, nonstoichiometric semiconductor nanocrystals for the realization of wide-spectrum-driven photocatalysis is highlighted.

Localized surface plasmon resonance (LSPR) is a unique optical property that has been extensively studied in noble metal, such as Au, Ag and Cu nanostructures[1–3]. Au nanostructures have attracted particularly significant attention because of the possibility of tuning the LSPR absorption across the visible to near infrared (NIR) spectral region. Such absorption tunability has substantial implications for utilizing the NIR spectrum, for which conventional semiconductor materials offer only limited choices. Recently, self-doped semiconductors, e.g., p-type Cu$_{2-x}$S, Cu$_{2-x}$Se and Cu$_{2-x}$Te, and n-type WO$_{3-x}$ and MoO$_{3-x}$, have demonstrated significant LSPR features primarily in

the NIR region[4,5]. The existence of intrinsic dopants generates a high density of free charge carriers for these nonstoichiometric semiconductors, inducing LSPR in the relatively low-energy NIR region. In contrast to the LSPR in metals, which is attributed to the oscillation of free electrons, the LSPR in self-doped semiconductors arises from free carriers provided by intrinsic vacancies associated with nonstoichiometry. For example, the Cu vacancies of Cu$_{2-x}$S induce the formation of generous holes, while the O vacancies of WO$_{3-x}$ produce plenteous electrons. By changing the degree of doping, the LSPR frequency of the self-doped semiconductors can be tailored, which is not

A full list of affiliations appears at the end of the paper. ✉e-mail: chang.m.aa@m.titech.ac.jp; yclo@nycu.edu.tw; lin.yg@nsrrc.org.tw; diau@nycu.edu.tw; yhsu@nycu.edu.tw; yhsu@cc.nctu.edu.tw

possible to achieve in noble metals. As demonstrated by Luther et al.[6] and Kriegel et al.[7], the LSPR bands of p-type $Cu_{2-x}S$ and $Cu_{2-x}Se$ nanocrystals progressively grow in intensity while undergoing blue spectral shifts as the density of Cu vacancies increases; control over the Cu vacancy density, *i.e.*, the x value, further allows dynamic regulation of the LSPR frequency, thus offering the possibility of harnessing the entire NIR spectrum. Similarly, the concentration of O vacancies in n-type $WO_{3-x}$[8,9] and $MoO_{3-x}$[10–13] can also be tuned to manipulate the LSPR bands, rendering plasmonic n-type semiconductors suitable for filling the gap in harvesting the NIR spectrum.

Solar hydrogen fuel has sparked substantial interest over the past half century, as it has the potential to meet the growing global energy demand. The utilization of solar energy to produce hydrogen over semiconductor photocatalysts has realized the core concept of sustainable energy development. The upper-limit of the solar-to-hydrogen conversion efficiency is governed by the light absorption capability of the photocatalysts. Extending the light absorption range to enhance the photon harvesting capacity is therefore indispensable for maximization of the photocatalytic activity. Note that the energy distribution of solar light is approximately 6.8% in the UV ($\lambda < 400$ nm), 38.9% in the visible ($\lambda = 400-700$ nm) and 54.3% in the NIR ($\lambda = 700-3000$ nm) ranges. The photons produced from NIR irradiation with a wavelength longer than 1000 nm represent a vast source of untapped energy. Most photocatalysts developed thus far are only capable of harvesting the solar spectrum in the UV and visible ranges. There are few choices among the currently available photocatalysts that can respond to NIR irradiation. The creation of NIR-responsive photocatalysts has therefore been seen as a prerequisite for realizing wide-spectrum-driven hydrogen production. Conventional NIR-responsive photocatalysts are limited to certain narrow-bandgap semiconductors, such as lead[14] and mercury chalcogenides[15]. The high toxicity, poor stability and reduced redox power due to bandgap narrowing have restricted their practical utilization in solar hydrogen production. Au nanostructures, on the other hand, are a recent addition to the NIR-responsive photocatalyst toolbox[16,17]. Nevertheless, the low conversion efficiency of the plasmonic energy of Au nanostructures, resulting from the ultrafast relaxation and recombination of hot carriers[18,19], has hindered their utility as a photocatalyst in solar hydrogen production. Compared with Au nanostructures, self-doped semiconductors possess more tailorable plasmonic features that can be secured for promising solar hydrogen production. The features include feasible dynamic control over the LSPR response and a larger degree of LSPR wavelength extension. Especially for $Cu_{2-x}S$, the achievable LSPR wavelength can range from 700 nm to beyond 2000 nm, nearly spanning the entire NIR spectrum. In concert with its moderate bandgap, $Cu_{2-x}S$ can be responsive to both visible and NIR irradiation, harnessing more than 90% of the total solar irradiance.

In contrast to the numerous reports dealing with photocatalytic hydrogen production over plasmonic Au, the number of reports tackling solar hydrogen production over plasmonic $Cu_{2-x}S$ is still very limited[20,21]. In this work, $Au@Cu_7S_4$ yolk@shell nanocrystals were synthesized and employed as photocatalysts for remarkable hydrogen production across the visible to NIR spectral region. The yolk@shell nanostructures possess many fascinating material properties suitable for photocatalytic reactions. First, the yolk particles are encapsulated in the shell, preventing their aggregation and detachment during the reaction process and thus ensuring superior long-term stability. Second, the hollow shell provides abundant active sites by offering both inner and outer surfaces for accessing the reacting species. Third, the permeable shell allows diffusion of reacting species; the void space within the shell can thus provide a confined space that can function as a robust nanoreactor to expedite the interactions of reactant and product species. Furthermore, the mobile yolk particles can stir the reaction solution to create a homogeneous environment, which accelerates the mass transport kinetics to increase the reaction rate as

well. Analytical results of in-situ X-ray absorption spectroscopy (XAS) and ultrafast transient absorption spectroscopy (TAS) validate the proposed vectorial charge transfer mechanism. Combined with the advantageous features of yolk@shell nanostructures, $Au@Cu_7S_4$ achieves a peak quantum yield (AQY) of 9.4% at 500 nm and a record-breaking AQY of 7.3% at 2200 nm for hydrogen production in the absence of additional co-catalysts.

## Results and discussion
### Microstructural investigations
The synthesis of $Au@Cu_7S_4$ required conducting of a sulfidation reaction on the $Au@Cu_2O$ core@shell nanocrystal template. $Au@Cu_2O$ was prepared by using a chemical reduction method[22–26]. A further sulfidation reaction can transform $Cu_2O$ into $Cu_7S_4$. Because of the nanosized Kirkendall effect[27], transformation from $Cu_2O$ into $Cu_7S_4$ was accompanied by the formation of abundant voids and their coalescence into an entire hollow space. Consequently, $Au@Cu_7S_4$ comprising a movable Au particle surrounded by a hollow $Cu_7S_4$ shell was formed. The resultant $Au@Cu_7S_4$ was first characterized by transmission electron microscopy (TEM) to visualize the microstructural features. As displayed in Fig. 1a-c, $Au@Cu_7S_4$ possessed a yolk@shell nanostructure, in which an individual particle was encapsulated in a hollow shell. Note that the yolk nanoparticles were randomly distributed within the shell, suggesting that they can move freely inside the hollow shell. The movement of the Au yolk inside the $Cu_7S_4$ shell can be witnessed by real-time TEM observations[26,28]. The results of high-resolution TEM (HRTEM, Fig. 1g), selected area electron diffraction (SAED, Fig. 1i), energy-dispersive X-ray spectrometry (EDS, Fig. 1j) and X-ray diffraction (XRD, Supplementary Fig. 1a) analysis further confirmed the compositions of the yolk particles as fcc Au and the shell as monoclinic $Cu_7S_4$. In this study, three Au contents were employed to produce $Au@Cu_7S_4$ with gradually decreasing void sizes. As determined from Fig. 1a-c, the void sizes were $65.7 \pm 5.6$ nm, $40.0 \pm 4.6$ nm and $26.5 \pm 3.0$ nm for $1\text{-}Au@Cu_7S_4$, $3\text{-}Au@Cu_7S_4$ and $5\text{-}Au@Cu_7S_4$, respectively. The controllability of the void size enabled us to explore the influence of the void size on the photocatalytic efficiency of $Au@Cu_7S_4$. By carefully controlling the experimental conditions associated with the sulfidation reaction, the shell thickness of the three $Au@Cu_7S_4$ can be adjusted to a fixed value of approximately $11.7 \pm 1.5$ nm. This adjustment allowed the exclusion of the influence of the shell thickness on the photocatalytic properties. On the other hand, the particle size distribution of Au for the three $Au@Cu_7S_4$ was also examined. The size of the Au yolk of $1\text{-}Au@Cu_7S_4$, $3\text{-}Au@Cu_7S_4$ and $5\text{-}Au@Cu_7S_4$ was $15.3 \pm 0.8$ nm, $15.2 \pm 0.8$ nm and $15.2 \pm 0.6$ nm, respectively. The consistency in Au size distribution also excluded its influence on the photocatalytic properties. For comparison, pure $Cu_7S_4$, pure Au and a physical mixture of pure Au and pure $Cu_7S_4$ (denoted as $Au+Cu_7S_4$) were also prepared and characterized in Fig. 1d–f. Pure $Cu_7S_4$ shared similar hollow structural features with $Au@Cu_7S_4$ except for the absence of the encapsulated Au particles. The void size and shell thickness of pure $Cu_7S_4$ were $47.1 \pm 9.2$ nm and $11.5 \pm 2.3$ nm, respectively. Pure Au particles had a uniform particle size of $15.2 \pm 1.0$ nm. $Au+Cu_7S_4$, on the other hand, was characterized by agglomeration of Au particles at the outer surface of hollow $Cu_7S_4$.

### In-situ XAS and charge transfer dynamics
The optical properties and band structure of $Au@Cu_7S_4$ were investigated with absorption, photoluminescence (PL) and ultraviolet photoelectron spectroscopy (UPS). According to the analytical results illustrated in Supplementary information, a plausible band alignment for $Au@Cu_7S_4$ to interpret the interfacial charge transfer pathways is depicted in Supplementary Fig. 1d. For $Au@Cu_7S_4$, as $Cu_7S_4$ and Au were brought in contact, the lower $E_F$ of Au induced an upward bending of the band edge in $Cu_7S_4$ as a result of the depletion of electrons. Upon band edge excitation, the upward band bending at the interface

facilitated photoexcited hole transfer from $Cu_7S_4$ to Au and enabled the photogenerated electrons to be concentrated in $Cu_7S_4$. Because the photoexcited holes were separated from the photogenerated electrons, radiative electron-hole recombination could be reduced to cause a depressed PL intensity for $Au@Cu_7S_4$. The interfacial upward band bending can also steer the dynamics of hot charge carriers produced from plasmonic Au and plasmonic $Cu_7S_4$. An instance of this scenario would be the injection of hot electrons from Au into $Cu_7S_4$ along the bent conduction band (CB) and the pass of hot holes from $Cu_7S_4$ to Au through the bent valence band (VB). To validate the charge transfer mechanism for these hot carriers, in-situ XAS measurements were conducted on $Au@Cu_7S_4$ by introducing a secondary irradiation to excite the LSPR. The experiment separately utilized irradiation of AM 1.5 G, visible and NIR light to explore the pathways of charge transfer associated with plasmonic Au and plasmonic $Cu_7S_4$. The real-time evolution of the unoccupied density of states (UDOS) with light irradiation can be derived by comparing the XAS spectra of Cu and Au obtained under irradiation and dark conditions[29–31]. Fig. 2a, b show the Cu $L_3$-edge and Au $L_3$-edge spectra for $Au@Cu_7S_4$ recorded in the dark and under the three irradiation conditions. At the Cu $L_3$-edge, three distinct peaks were detected at approximately 931.9 ($a_1$), 934.9 ($a_2$) and 932.8 eV ($a_3$). The prominent peak, $a_1$, was associated with the electron excitation from Cu $2p$ to the unoccupied $3d$ state that constituted the CB edge of $Cu_7S_4$. The considerably broad peaks, $a_2$ and $a_3$, were ascribed to the electronic transition from Cu $2p$ to the mixed $s$ and $d$ empty states and from Cu $2p$ to the empty ligand states of the CB of

$Cu_7S_4$, respectively. Further implications in the charge transfer pathways can be obtained by analyzing the Cu $L_3$-edge spectra under different irradiation conditions. Under AM 1.5 G irradiation, a perceivable decrease in spectral intensity was observed, which can be reflected by the negative intensity difference ($\Delta A_1 = A_{AM\ 1.5\ G} - A_{dark}$) distributed in the three peak regions in Fig. 2a. The decrease in the XAS intensity signified a reduced UDOS for the $Cu_7S_4$ component of $Au@Cu_7S_4$ as a result of the accumulation of excited electrons. Note that AM 1.5 G irradiation not only caused band edge excitation of $Cu_7S_4$, but also induced plasmonic excitation of both Au and $Cu_7S_4$, as derived from the band alignment illustrated in Supplementary Fig. 1d. Consequently, the photoexcited electrons and holes from $Cu_7S_4$ can accompany the hot electrons from plasmonic Au and the hot holes from plasmonic $Cu_7S_4$ participating in the charge relaxation processes. As depicted in Fig. 2c, the possible charge transfer pathways included photoexcited hole transfer from $Cu_7S_4$ to Au, hot electron injection from plasmonic Au to $Cu_7S_4$ and hot hole injection from plasmonic $Cu_7S_4$ to Au. These events can operate together to result in an accumulation of excited electrons at $Cu_7S_4$, leading to a decrease in UDOS for $Au@Cu_7S_4$, as evidenced by a negative $\Delta A_1$. In order to decouple these charge transfer pathways, spectral comparison between dark condition and visible ($\lambda = 400–700$ nm) or NIR ($\lambda > 800$ nm) irradiation was carried out. Even with visible irradiation, the reduction in the intensity of the Cu $L_3$-edge spectrum remained noticeable, as reflected by a considerably negative $\Delta A_2$ in the three peak regions. Note that visible irradiation merely caused two types of excitations, namely band edge excitation of $Cu_7S_4$

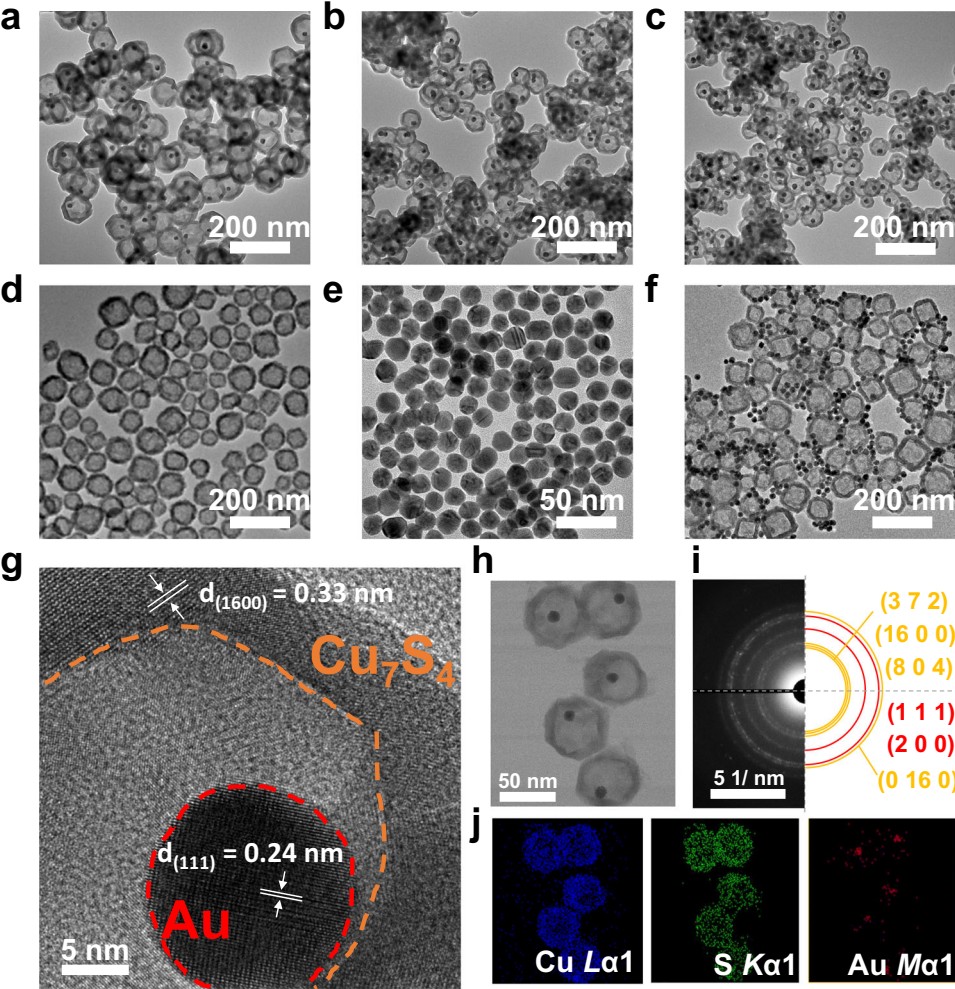

**Fig. 1 | Microstructural features of $Au@Cu_7S_4$.** TEM images of **a** 1-$Au@Cu_7S_4$, **b** 3-$Au@Cu_7S_4$, **c** 5-$Au@Cu_7S_4$, **d** pure $Cu_7S_4$, **e** pure Au, **f** Au+$Cu_7S_4$. **g** HRTEM image of 5-$Au@Cu_7S_4$. **h** TEM image and corresponding **i** SAED pattern, **j** TEM-EDS mapping profiles.

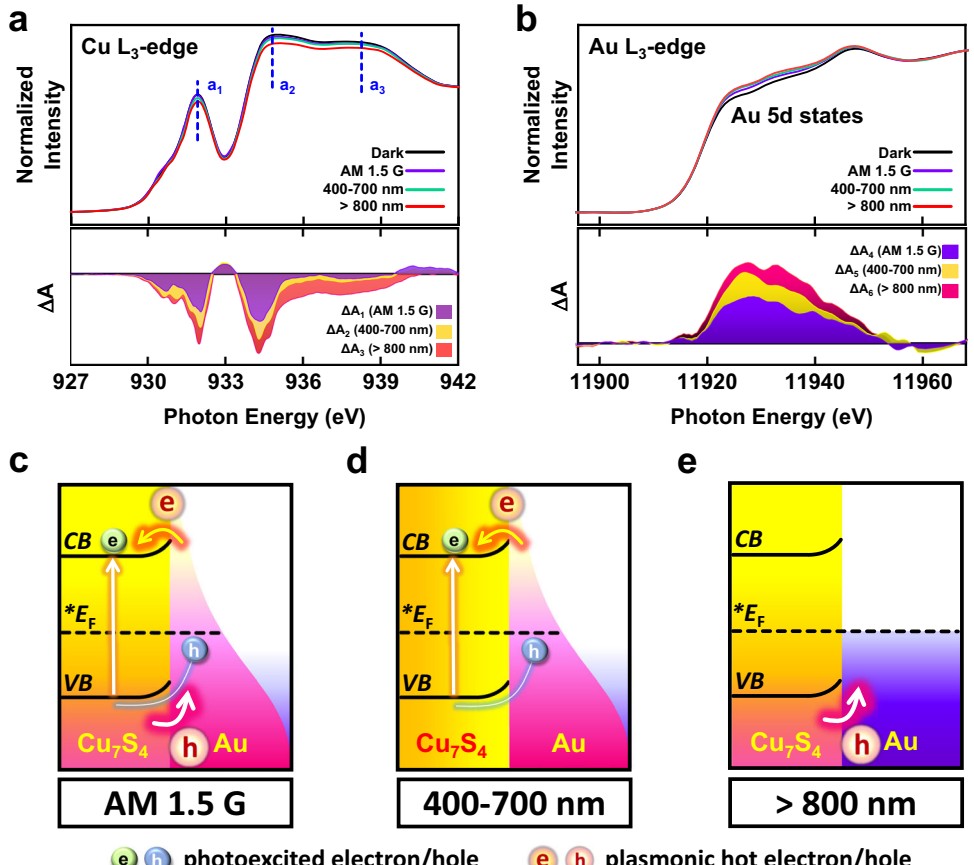

**Fig. 2 | In-situ XAS. a** Cu $L_3$-edge spectra and corresponding intensity difference spectra (ΔA) and **b** Au $L_3$-edge spectra and corresponding intensity difference spectra (ΔA) for 5-Au@Cu$_7$S$_4$ recorded under dark and three irradiation conditions. Schematic illustration of charge transfer scenarios for Au@Cu$_7$S$_4$ under **c** AM 1.5 G irradiation, **d** visible irradiation, **e** NIR irradiation.

and plasmonic excitation of Au. While the photoexcited holes of Cu$_7$S$_4$ were transported to Au, the photogenerated electrons were retained in Cu$_7$S$_4$, and the hot electrons produced from plasmonic Au were injected into Cu$_7$S$_4$. Figure 2d depicts how these two events can occur collaboratively, reducing the UDOS for Au@Cu$_7$S$_4$ to give a negative ΔA$_2$. Notably, the obtained ΔA$_2$ was more negative than ΔA$_1$, suggesting that visible irradiation secured a greater accumulation of excited electrons at Au@Cu$_7$S$_4$ than AM 1.5 G irradiation. It is necessary to mention that visible irradiation was provided by placing a bandpass filter (400–700 nm) over AM 1.5 G irradiation to extract the visible photons from the AM 1.5 G spectrum, giving an irradiation intensity that was 48.2% of the initial AM 1.5 G irradiation. This would enable a quantitative comparison of UDOS between the results from AM 1.5 G and visible irradiations. The observed greater accumulation of excited electrons at Cu$_7$S$_4$ under visible irradiation thus reflected a factual situation caused by the visible photons of the AM 1.5 G spectrum. This implied that the joint operation of the three excitations under AM 1.5 G irradiation was less efficient in inducing excited electron accumulation for Au@Cu$_7$S$_4$ compared to the collaborative operation of the two excitations under visible irradiation. The possible reason lies in the interference of charge transfer between hot electrons and hot holes under AM 1.5 G irradiation. As the hot electrons of Au were injected into Cu$_7$S$_4$, they might encounter the hot holes of Cu$_7$S$_4$ to undergo electron-hole recombination. This backward charge recombination would reduce the number of excited electrons accumulated at Cu$_7$S$_4$, which could account for the observed less negative ΔA$_1$ than ΔA$_2$. Further comparison with the Cu $L_3$-edge spectrum of Au@Cu$_7$S$_4$ under NIR excitation enabled the exclusive examination on the transfer of hot holes produced from plasmonic Cu$_7$S$_4$. The difference spectrum revealed an even more negative ΔA$_3$,

suggesting that NIR irradiation can bring forth much more excited electron accumulation at Au@Cu$_7$S$_4$ as a result of the hot hole transfer to Au (Fig. 2e). Similarly to the concept of applying visible irradiation, NIR irradiation was provided by placing a long-pass filter (>800 nm) over AM 1.5 G irradiation to extract the NIR photons from the AM 1.5 G spectrum, giving an irradiation intensity that was 54.6% of the initial AM 1.5 G irradiation. This would again enable a quantitative comparison of UDOS between the results from AM 1.5 G and NIR irradiations. These comparative results corroborated that hot hole transfer can interfere with hot electron transfer under AM 1.5 G illumination. Even though hot hole transfer was intrinsically efficient, it could compromise with hot electron transfer on the accumulation of excited electrons, leading to the least reduced UDOS observed under AM 1.5 G irradiation. It might be argued that the strong light absorption of Au@Cu$_7$S$_4$ at visible and NIR regions may directly lead to the enlarged extent of UDOS change observed under visible and NIR irradiations. This argument can be ruled out according to the following considerations. For the in-situ XAS measurements, the visible and NIR irradiations were provided by placing filters over AM 1.5 G irradiation to separately extract the visible and NIR photons from the AM 1.5 G spectrum. The intensity of visible and NIR irradiations was therefore substantially lower than that of AM 1.5 G illumination. Despite the much lower irradiation intensity, visible and NIR irradiations still caused a larger extent of UDOS decrease, disclosing the lower efficiency of inducing excited electron accumulation as a result of the interference of the three excitations under AM 1.5 G irradiation.

Additionally, the Au $L_3$-edge profiles of Au@Cu$_7$S$_4$ were examined in Fig. 2b to offer supplementary insights into the proposed charge transfer scenarios. The profiles exhibited three unambiguous bands in

the energy range from 11900 to 11980 eV, which were believed to result from the $2p_{3/2,1/2}$ to $5d$ dipole transitions[32]. Under AM 1.5 G irradiation, the difference spectra showed a positive intensity difference $\Delta A_4$ in the three band regions, reflecting an elevation in the UDOS for the Au component of $Au@Cu_7S_4$. This characteristic complemented the decreased UDOS of the $Cu_7S_4$ component derived from the comparison of Cu $L_3$-edge spectra under AM 1.5 G irradiation, validating that excited electrons were accumulated at the $Cu_7S_4$ of $Au@Cu_7S_4$ as a result of the joint operation of the three excitations. A consistent result can be obtained from the spectral comparison of the Au $L_3$-edge under visible irradiation, showing a more positive $\Delta A_5$ than $\Delta A_4$. This outcome strongly supported the assertion that AM 1.5 G irradiation was less effective than visible irradiation in providing $Au@Cu_7S_4$ with accumulated excited electrons due to the interference of charge recombination between hot electrons and hot holes. More importantly, the most positive $\Delta A_6$ recorded under NIR irradiation disclosed the intrinsically efficient hot hole transfer from $Cu_7S_4$ to Au in dictating excited electron accumulation for $Au@Cu_7S_4$. This result has significant implications in harvesting NIR energy for solar hydrogen production by exploiting the LSPR of $Au@Cu_7S_4$.

To depict a comprehensive picture of interfacial charge transfer processes, the carrier relaxation dynamics of the samples were further studied by using visible-NIR pump-probe TAS in the time region spanning from 100 fs to 300 μs with probing wavelengths between 550 and 2000 nm. The charge transfer dynamics of the $Cu_7S_4$ component for pure $Cu_7S_4$ and $Au@Cu_7S_4$ were exclusively monitored and compared because $Cu_7S_4$ was mainly responsible for harvesting incident photons for charge carrier generation. For the TAS experiments, femtosecond (for fs-ns time spans) or nanosecond (for ns-μs time spans) visible and NIR pulses were employed to render $Cu_7S_4$ component band edge or plasmonic excitation. The femtosecond TAS profiles of pure $Cu_7S_4$ and $5$-$Au@Cu_7S_4$ are shown in Supplementary Figs. 2a, d and Supplementary Figs. 2b, e, respectively. The samples were pumped using 500 and 1400 nm excitation pulses resonant with excitonic and plasmonic absorptions, respectively, and probed in the sub-band gap region from 550 to 950 nm to uncover the relaxation dynamics of the surface trap states. Both pure $Cu_7S_4$ and $5$-$Au@Cu_7S_4$ showed a rapidly decaying photoinduced absorption (PIA) band due to hot hole injection into the surface states, irrespective of the excitation wavelength. Although the spectral features for pure $Cu_7S_4$ and $5$-$Au@Cu_7S_4$ were similar, their magnitudes and decay dynamics substantially varied, indicative of the divergence in the underlying charge transfer mechanism. Under the same excitation conditions, $5$-$Au@Cu_7S_4$ showed stronger PIA bands due to its larger absorption cross-section. It is important to mention that plasmonic excitation by NIR pulses could produce PIA bands with similar magnitudes to those produced under band edge excitation despite a much weaker laser fluence of NIR pulses (7 times smaller than the visible pulses). This outcome suggested that the hot hole dynamics of $Cu_7S_4$ upon plasmonic excitation were as effective as the charge carrier dynamics of $Cu_7S_4$ from band edge excitation. The carrier thermalization times were observed to occur between 0.6 and 0.8 ps, as displayed in Supplementary Figs. 2c, f, with $5$-$Au@Cu_7S_4$ showing a slightly faster cooling rate. Nanosecond TAS experiments were further performed to probe the surface state-mediated carrier relaxation dynamics and thereby attain a better understanding of the carrier relaxation mechanism. Figure 3 presents the spectral and temporal profiles of NIR transients (850 to 2500 nm) for pure $Cu_7S_4$ and $5$-$Au@Cu_7S_4$ obtained by pumping the samples using nanosecond visible (532 nm) and NIR (1064 nm) pulses. The NIR transients of pure $Cu_7S_4$ comprised weak spectral features and exhibited instrument response limited decays under both visible and NIR excitations, whereas $5$-$Au@Cu_7S_4$ showed significant bleaching of plasmonic bands with retarded recombination processes. The significant photo-bleaching of the plasmonic bands of $5$-$Au@Cu_7S_4$ under visible excitation was attributable to the increase in

absorptivity, similar to the cause of the strong PIA bands observed in femtosecond TAS profiles. As Fig. 3c, f compare, the dramatic difference in the recombination dynamics between pure $Cu_7S_4$ and $5$-$Au@Cu_7S_4$ suggested that distinct recombination pathways existed. One possible cause for the delayed recombination for $5$-$Au@Cu_7S_4$ could be the creation of charge separation states at the interface between $Cu_7S_4$ and Au. Such charge separation states might involve direct electron transfer as well as resonance energy transfer between metal and semiconductor[33].

Based on the spectral comparison results, a plausible mechanism to account for the carrier relaxation pathways for pure $Cu_7S_4$ and $5$-$Au@Cu_7S_4$ is proposed in Fig. 3g. Upon visible excitation, photoexcited charge carriers were produced in the VB, CB and surface plasmonic band (SPB), depleting both excitonic and plasmonic states. The produced carriers thermalized within 0.8 ps for pure $Cu_7S_4$ and 0.7 ps for $5$-$Au@Cu_7S_4$ via carrier-carrier scattering, electron-phonon scattering and/or electron-plasmon scattering routes. Meanwhile, the thermalized carriers could populate the surface states of $Cu_7S_4$ created by the Cu vacancies, followed by recombination with the delocalized electrons from the CB. The surface states could be categorized into shallow and deep trap states based on the observed multi-exponential relaxation dynamics of the samples. As derived from Fig. 3c, pure $Cu_7S_4$ showed biexponential recombination kinetics with two time constants of 0.1 and 0.7 μs. Noticeably, the surface state-mediated relaxation was greatly retarded for $5$-$Au@Cu_7S_4$, exhibiting three time constants of 2, 10, and 160 μs. The retarded relaxation kinetics of $5$-$Au@Cu_7S_4$ were believed to originate from the transfer of the trapped holes from $Cu_7S_4$ to Au. Under NIR excitation, hot carriers associated with the SPB of the $Cu_7S_4$ component were produced in both pure $Cu_7S_4$ and $5$-$Au@Cu_7S_4$, which thermalized with time constants of 0.7 and 0.6 ps, respectively. For pure $Cu_7S_4$, the thermalized hot carriers rapidly recombined within 5 ns (approximately equal to the pulse limited response). For $5$-$Au@Cu_7S_4$, an appreciable fraction of additional delayed recombination (6 μs) was observed (Fig. 3f), evidencing the existence of charge separation states induced by hot hole transfer to Au. The results of TAS analysis revealed the prevalence of long-lived charge separation states for $Au@Cu_7S_4$ under both visible and NIR excitation, which may prolong the lifetime of the delocalized electrons to facilitate hydrogen production. In order to investigate the influence of void size on the charge transfer dynamics for $Au@Cu_7S_4$, the TAS profiles of the three $Au@Cu_7S_4$ are further compared in Supplementary Fig. 3. As tabulated in Supplementary Tables 1 and 2, the transient kinetics of the three $Au@Cu_7S_4$ did not show appreciable changes in the decay rates under the two excitation conditions. This outcome signified that the void size did not have a significant effect on the charge transfer dynamics of $Au@Cu_7S_4$.

## Activity for solar hydrogen production

The comparative results of solar hydrogen production are displayed in Fig. 4a. There was nearly no hydrogen produced from pure Au, suggesting that Au colloids were inactive toward solar hydrogen production. Pure $Cu_7S_4$, on the other hand, showed a modicum of activity, reaching a hydrogen production rate of 25.1 μmol $h^{-1}$ $g^{-1}$. Compared to pure $Cu_7S_4$, the three $Au@Cu_7S_4$ all displayed an increased hydrogen production rate, disclosing the beneficial function of the Au yolk for enhancing the photocatalytic performance of $Cu_7S_4$. As depicted in Fig. 3g, the yolk Au particles can function as a charge separation enhancer for $Cu_7S_4$ under both visible and NIR excitations, which can increase the total number of available charge carriers to facilitate hydrogen production. For the three $Au@Cu_7S_4$, the hydrogen production rate increased with decreasing void size, with $5$-$Au@Cu_7S_4$ showing the highest hydrogen production rate of 211.0 μmol $h^{-1}$ $g^{-1}$. Note that the decrease in the void size did not have a significant effect on the charge transfer dynamics for $Au@Cu_7S_4$, as demonstrated from the TAS analytical results in Supplementary Fig. 3. The influence of

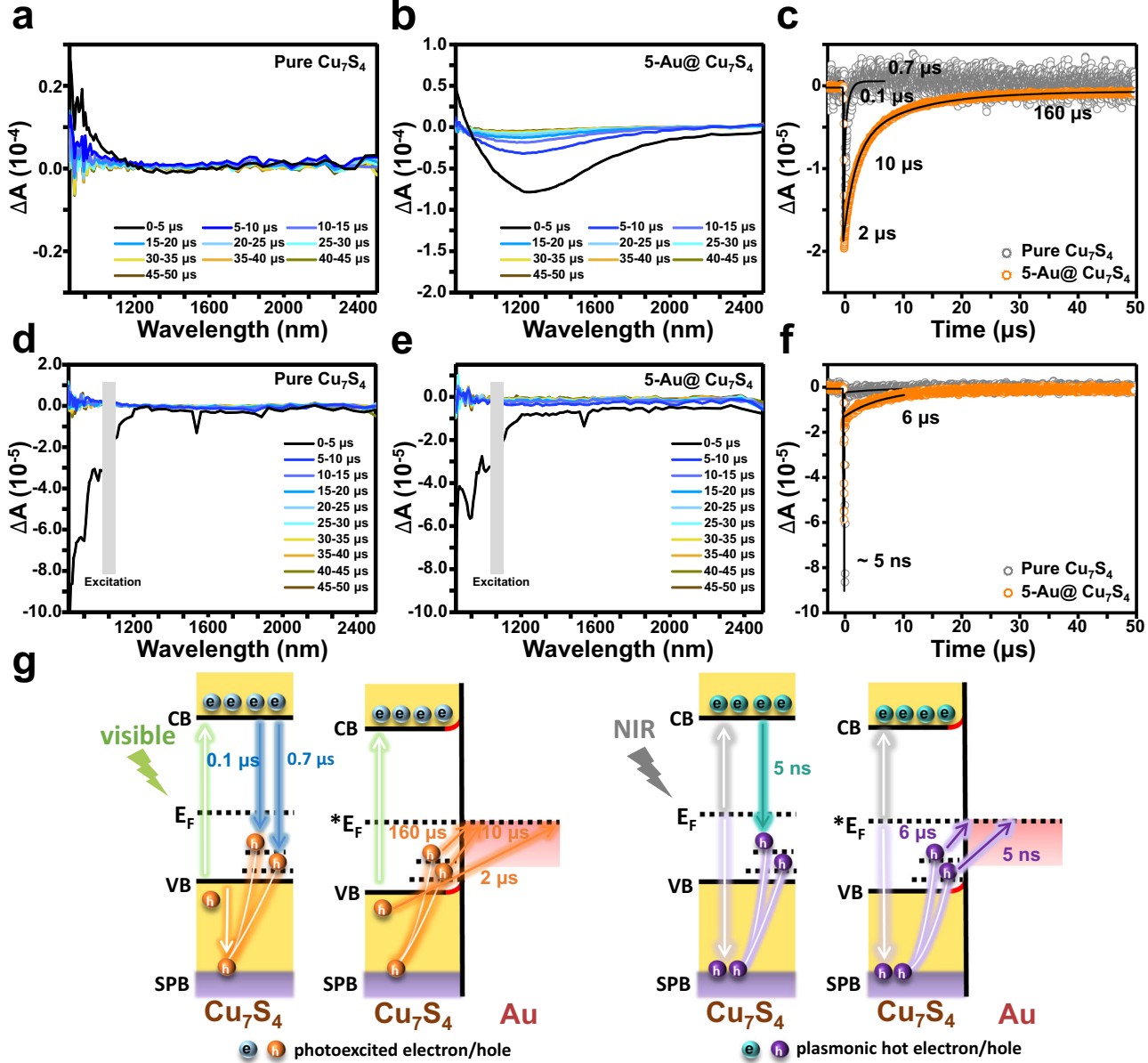

**Fig. 3 | Nanosecond TAS.** Spectral and temporal profiles for pure $Cu_7S_4$ and 5-Au@$Cu_7S_4$ measured by using excitation wavelengths of **a–c** 532 nm and **d–f** 1064 nm. **c** and **f** were obtained by integrating the area under the intensity curves from 900 to 2400 nm. **g** Proposed mechanism of carrier relaxation pathways for pure $Cu_7S_4$ and Au@$Cu_7S_4$ under visible and NIR excitation conditions.

charge transfer dynamics on the varied photocatalytic activities of the three Au@$Cu_7S_4$ can therefore be ruled out. On the other hand, Au@$Cu_7S_4$ with a smaller void size was expected to provide more accessible $Cu_7S_4$ surfaces and thus offer a larger number of active sites. According to the Brunauer-Emmett-Teller (BET) analytical results, the specific surface areas of 1-Au@$Cu_7S_4$, 3-Au@$Cu_7S_4$ and 5-Au@$Cu_7S_4$ were 15.35, 16.87 and 19.79 $m^2\ g^{-1}$, respectively. By normalizing the hydrogen production rate to the specific surface area, the specific hydrogen production activities could be further obtained, giving 5.5, 7.5 and 10.7 $\mu mol\ h^{-1}\ m^{-2}$ for 1-Au@$Cu_7S_4$, 3-Au@$Cu_7S_4$ and 5-Au@$Cu_7S_4$, respectively. The increase in specific hydrogen production activity with decreasing void size was still noticeable, disclosing that there existed other predominant factors dictating the much enhanced performance for 5-Au@$Cu_7S_4$. Since the Au yolk was mobile inside the $Cu_7S_4$ shell, Au could constantly stir the electrolyte inside the hollow $Cu_7S_4$ to create a homogeneous environment. For 5-Au@$Cu_7S_4$ with the smallest void size, the mobile Au can stir the

reaction solution much vigorously as a result of the much confined space between Au and $Cu_7S_4$. This would accelerate the reaction kinetics of hydrogen production to maximize the photocatalytic performance. Because diffusion of hydrogen across the shell was necessary for maintaining sustained hydrogen production, the mass transport kinetics across the shell should also be considered. To resolve this issue, release experiments using rhodamine B (RhB) as an optical probe were performed. By monitoring the temporal release profiles of RhB over Au@$Cu_7S_4$, the mass transport kinetics across the $Cu_7S_4$ shell can be estimated. The comparative results for the three Au@$Cu_7S_4$ are shown in Supplementary Fig. 4, in which the RhB release rate was found to increase with decreasing void size. By fitting these data to an empirical equation (see details in Methods section)[34,35], the diffusion coefficient (D) across the $Cu_7S_4$ shell can be further calculated, giving $2.35 \times 10^{-20}$, $1.14 \times 10^{-19}$ and $1.27 \times 10^{-19}\ m^2\ s^{-1}$ for 1-Au@$Cu_7S_4$, 3-Au@$Cu_7S_4$ and 5-Au@$Cu_7S_4$, respectively. Among the three Au@$Cu_7S_4$, 5-Au@$Cu_7S_4$ showed the highest D. As the void size

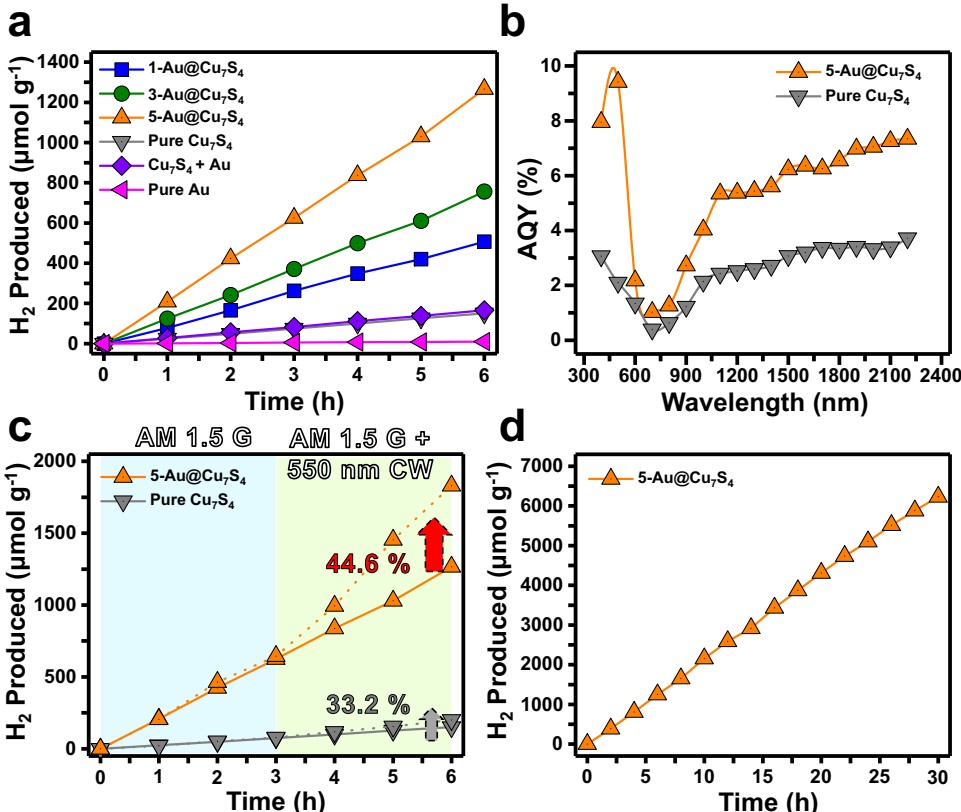

**Fig. 4 | Activity for solar hydrogen production. a** Comparison of hydrogen production activity on six relevant samples. **b** AQY values measured under different incident wavelengths for pure $Cu_7S_4$ and 5-$Au@Cu_7S_4$. **c** Hydrogen production activity on pure $Cu_7S_4$ and 5-$Au@Cu_7S_4$ with and without additional visible irradiation. **d** Extended use of 5-$Au@Cu_7S_4$ in solar hydrogen production for 30 successive hours.

was reduced, the concentration gradient of RhB between the interior of $Au@Cu_7S_4$ and the bulk of the surrounding solution was greatly enlarged. This could explain why a reduced void size can expedite the diffusion of RhB across the $Cu_7S_4$ shell. The results of RhB release experiments reflected that the mass transport kinetics across the shell could be facilitated as the void size was reduced, which was considered to be another reason for the much enhanced hydrogen production activity for 5-$Au@Cu_7S_4$.

The critical role of the yolk@shell nanostructures in achieving superior hydrogen production was further evidenced by assessing the performance of Au+$Cu_7S_4$. The hydrogen production activity of Au+$Cu_7S_4$ was 27.6 μmol h$^{-1}$ g$^{-1}$, approximating the performance of pure $Cu_7S_4$. As Fig. 1f displays, Au+$Cu_7S_4$ was characterized by noticeable surface coverage of $Cu_7S_4$ by aggregated Au. This microstructural disorder can largely deteriorate the effectiveness of interfacial charge transfer, causing inferior activity. Note that Au+$Cu_7S_4$ was obtained by simply mixing Au colloids and pure $Cu_7S_4$ suspension with their concentrations deliberately adjusted to equate those of 5-$Au@Cu_7S_4$. In other words, the amount of Au and $Cu_7S_4$ in Au+$Cu_7S_4$ was respectively equal to the amount of Au and $Cu_7S_4$ in 5-$Au@Cu_7S_4$. With this deliberate control can the activity performance comparison of Au+$Cu_7S_4$ with $Au@Cu_7S_4$ give a reliable conclusion that yolk@shell nanostructures played a critical role in achieving superior hydrogen production. To elucidate the cause behind the enhanced activity of $Au@Cu_7S_4$, the AQYs of hydrogen production were measured under irradiation with monochromatic light. As shown in Fig. 4b, pure $Cu_7S_4$ and 5-$Au@Cu_7S_4$ exhibited similar spectral features with prominent AQYs across two distinct wavelength regions. For pure $Cu_7S_4$, the protruding AQY at wavelengths less than 700 nm was related to the sub-gap transition and band edge excitation of $Cu_7S_4$, while the prosperous AQY in the 700–2200 nm region originated from the plasmonic excitation of $Cu_7S_4$.

Compared to pure $Cu_7S_4$, 5-$Au@Cu_7S_4$ displayed much enhanced AQY across these two wavelength regions. It is important to note that the AQYs under NIR irradiation attained by 5-$Au@Cu_7S_4$ (AQY = 7.3% at 2200 nm) were substantially higher than the values of most of the NIR-responsive photocatalysts ever reported (Supplementary Table 3). As noted in Supplementary Table 4, the achievable AQY of 5-$Au@Cu_7S_4$ under visible irradiation (AQY = 9.4% at 500 nm) was also comparable to that of other state-of-the-art sulfide-based visible light-responsive photocatalysts. Note that there existed many sulfides photocatalysts exhibiting high hydrogen production activities. It should be, however, pointed out that those efficient sulfides photocatalysts were merely responsive to visible light rather than NIR irradiation. There are few choices among the currently available photocatalysts that can respond to NIR irradiation. In comparison with the state-of-the-art NIR-responsive photocatalysts reported so far, the current $Au@Cu_7S_4$ exhibited a record-breaking quantum yield of 7.3% at 2200 nm for hydrogen production. On top of that, the hydrogen production activity of $Au@Cu_7S_4$ toward visible irradiation (AQY = 9.4% at 500 nm) was also comparable to that of the efficient sulfides photocatalysts ever reported. It is also important to note that the currently reported AQY was pristine without the aid of any co-catalysts. By introducing suitable co-catalysts, the activity of 5-$Au@Cu_7S_4$ for solar hydrogen production can be further elevated. We considered two possible causes to account for the observed AQY enhancement. As a charge separation enhancer, the Au yolk can improve the carrier utilization efficiency of $Cu_7S_4$ by promoting both photoexcited hole transfer and hot hole injection dynamics, resulting in a significant AQY enhancement across the two wavelength regions. As a plasmonic antenna, the Au yolk, on the other hand, can sensitize the neighboring $Cu_7S_4$ to the wavelength at which the LSPR of Au is located by means of hot electron injection. As estimated from the empirical equation derived from Mie theory, the theoretical LSPR

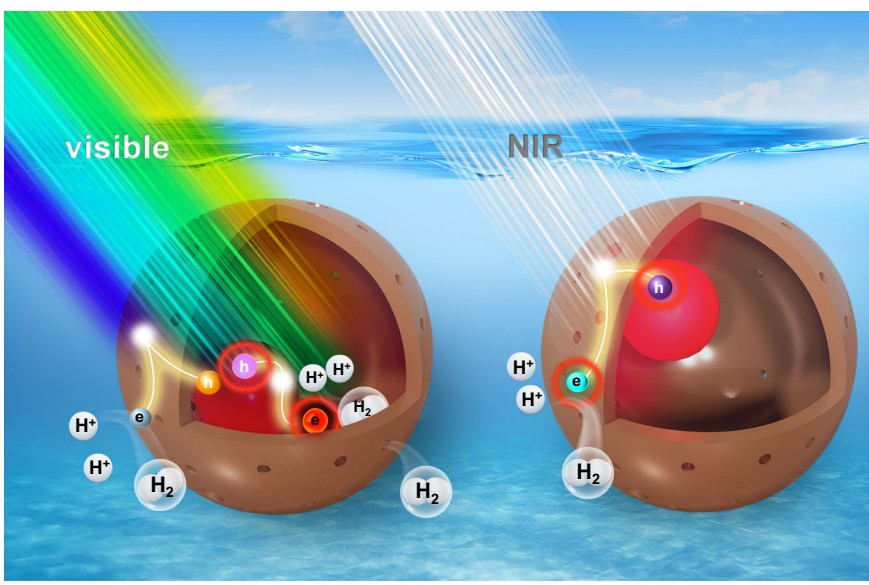

**Fig. 5 | Mechanism of solar hydrogen production.** Under visible irradiation, the photogenerated electrons concentrated at $Cu_7S_4$ can reduce protons to evolve hydrogen; the hot electrons from plasmonic Au can be injected into $Cu_7S_4$ and contribute to hydrogen production. Under NIR irradiation, the hot electrons delocalized at plasmonic $Cu_7S_4$ can reduce protons to produce hydrogen.

wavelength of Au for 5-Au@$Cu_7S_4$ was between 522.2 and 755.8 nm. It is, however, difficult to solely identify such an LSPR-induced AQY enhancement from Fig. 4b because this region spectrally overlapped with the AQY onset. To validate the plasmonic effect of Au, we further conducted solar hydrogen production measurements by introducing additional continuous-wave (CW) visible irradiation (550 nm, 48.7 μW cm$^{-2}$) that can excite the LSPR of Au. As illustrated in Fig. 4c, pure $Cu_7S_4$ exhibited a 32.2% increase in the hydrogen production rate under the additional visible irradiation, while 5-Au@$Cu_7S_4$ showed a 44.6% increase in activity under the same irradiation conditions. The activity increase for pure $Cu_7S_4$ was ascribed to the enhanced sub-gap absorption upon additional visible irradiation. By subtracting the contribution of enhanced sub-gap absorption, the activity enhancement due to the plasmonic effect of Au can be estimated for 5-Au@$Cu_7S_4$, which was approximately 12.4%.

In addition to the hydrogen production activity, the long-term stability is another key parameter to demonstrate the practicability of photocatalysts for solar hydrogen production. To evaluate this parameter, 5-Au@$Cu_7S_4$ was continuously used in solar hydrogen production for 30 successive hours. As shown in Fig. 4d, the recorded hydrogen production data exhibited a nearly straight line with reaction time, suggesting that 5-Au@$Cu_7S_4$ had considerably high stability. Furthermore, the discussions and relevant data showed in Supplementary information and Supplementary Figs. 5, 6 corroborated the high chemical and structural stability for Au@$Cu_7S_4$ toward solar hydrogen production. The high chemical and structural stability are especially important for demonstrating the practical use of chalcogenide photocatalysts, which typically have photocorrosion issues. The thermal effect induced by light irradiation has also been investigated and discussed in Supplementary information. Experimental results (Supplementary Figs. 7–17) revealed that the thermal effect on hydrogen production can be considered rather minor in the current system.

The density functional theory (DFT) calculations were performed to fundamentally understand the origin of the superiority of Au@$Cu_7S_4$ in hydrogen production. Since the real crystals of Au@$Cu_7S_4$ are large and highly complex, the structural models used for DFT calculations were simplified to obtain the convergence results. The optimized structures of pure Au, pure $Cu_7S_4$ and Au@$Cu_7S_4$ were

displayed in Supplementary Fig. 18, and the calculation details and results were shown in the Methods section and Supplementary Fig. 19. Gibbs free energy of hydrogen adsorption ($\Delta G_{H^*}$) on the three structural models was compared to look into the overall thermodynamics behaviors of hydrogen production. In general, the ideal catalysts suited for hydrogen production should have a $|\Delta G_{H^*}|$ value close to zero[36]. As summarized in Supplementary Fig. 19 and Supplementary Table 5, the computed $\Delta G_{H^*}$ value for Cu and S site of pure $Cu_7S_4$ was −5.17 and −5.05 eV, respectively, indicating that hydrogen was easy to adsorb on $Cu_7S_4$ to form strong $Cu_7S_4$-H* bonding. Since the $Cu_7S_4$-H* bonds were considerably stable, the subsequent desorption of H* would be suppressed to hinder hydrogen evolution[37]. In contrast, pure Au possessed an appropriate $\Delta G_{H^*}$ for the adsorption and desorption of H*; the relatively small $\Delta G_{H^*}$ (0.46 eV) suggested its promise as an intriguing catalyst for hydrogen production. The inadequacy for responding to light irradiation, however, limited the practice of pure Au in solar hydrogen production. Compared to pure $Cu_7S_4$, Au@$Cu_7S_4$ held a much lower $|\Delta G_{H^*}|$ value. The computed $\Delta G_{H^*}$ was 0.80 eV for Cu site and 0.23 eV for S site, signifying that both Cu and S sites of Au@$Cu_7S_4$ were thermodynamically favorable for proceeding with hydrogen evolution reaction. For Au@$Cu_7S_4$, the electronic interactions between Au and $Cu_7S_4$ may optimize hydrogen evolution by mediating hydrogen adsorption and desorption on $Cu_7S_4$[37–40]. This feature further suggested that $Cu_7S_4$ could be the active site for solar hydrogen production on Au@$Cu_7S_4$.

Based on the comparative results, a plausible mechanism to interpret the remarkable performance of Au@$Cu_7S_4$ in hydrogen production across the visible to NIR spectrum was conceived. As illustrated in Fig. 5, under visible irradiation, both band edge excitation of $Cu_7S_4$ and plasmonic excitation of Au occurred, producing photoexcited charge carriers and hot electrons, respectively. Owing to the upward band bending at the interface, the photoexcited holes of $Cu_7S_4$ would be favorably transported to Au. The photogenerated electrons would then be concentrated at $Cu_7S_4$ to achieve charge carrier separation. According to the results of DFT calculations, $Cu_7S_4$ was suggested to be the active site for hydrogen production on Au@$Cu_7S_4$. The photogenerated electrons concentrated at $Cu_7S_4$ can thus reduce protons and evolve hydrogen. As derived from the TAS data, the prevalence of long-lived charge separation states with a lifetime

component up to 160 μs can prolong the lifetime of the delocalized electrons at $Cu_7S_4$, which accounts for the substantially enhanced hydrogen production activity for $Au@Cu_7S_4$. The hot electrons produced from plasmonic Au, on the other hand, were highly energetic, which can overcome the barrier of the bent CB of $Cu_7S_4$ to be injected into $Cu_7S_4$[41]. These hot electrons contributed to hydrogen production as well. Under NIR irradiation, the hot holes generated at plasmonic $Cu_7S_4$ can preferentially transfer to Au. As the TAS analysis delivers, the retarded relaxation kinetics of hot holes with a lifetime of 6 μs can render extended survival of delocalized hot electrons, achieving noticeable hydrogen production as observed.

In summary, we have demonstrated the use of $Au@Cu_7S_4$ as dual-plasmonic photocatalysts for remarkable solar hydrogen production. The superiority of $Au@Cu_7S_4$ lies in the prevalence of long-lived charge separation states under both visible and NIR excitation along with the advantageous features of the yolk@shell nanostructures. The current study has delivered a conceptually attractive yet practically efficient dual-plasmonic photocatalyst paradigm capable of harvesting photons over the entire solar spectrum and beyond. It should be noted that dual-plasmonic heterostructures comprising plasmonic metals and plasmonic semiconductors have been widely investigated in recent years due to the intriguing optical properties resulting from the synergy of the two LSPR features[42,43]. Previous studies have demonstrated the extensive use of dual-plasmonic heterostructures in photothermal and biomedical applications. As discussed in Supplementary information, using dual-plasmonic heterostructures as photocatalysts in photocatalytic applications is still in its infancy. Our findings not only present a special type of plasmonic photocatalytic platform enabling solar fuel generation from untapped NIR energy, but also advance the fundamental understanding of peculiar nonstoichiometric semiconductor nanocrystals and their utility in photocatalysis. In particular, the revelation of the remarkable NIR activity of $Au@Cu_7S_4$ is exciting and inspiring because it can fill the gap in harvesting the NIR spectrum for the currently available photocatalysts.

## Methods
### Synthesis of $Au@Cu_7S_4$
A chemical reduction method was used to prepare the $Au@Cu_2O$ core@shell nanocrystal template[24,25]. First, Au colloids (0.25 mM) were synthesized using the citrate reduction method. A given volume of Au colloids (1.0, 3.0 or 5.0 mL) was then dispersed in a $CuSO_4$ solution (2.0 mL, 0.01 M) at 35 °C, followed by the addition of a NaOH solution (1.5 mL, 0.1 M) and an L-ascorbic acid solution (0.5 mL, 0.1 M). After this mixture was steadily stirred at 35 °C for 10 min, a suspension containing $Au@Cu_2O$ colloids was obtained. To conduct the sulfidation reaction, a $Na_2S$ solution (400 μL, 0.1 M) was injected into the $Au@Cu_2O$ suspension, leading to the transformation from $Au@Cu_2O$ core@shell nanocrystals into $Au@Cu_7S_4$ yolk@shell nanocrystals. In this work, the size of the void space in the resultant $Au@Cu_7S_4$ was controlled by adjusting the shell thickness of the initial $Au@Cu_2O$ template. In principle, a thinner $Cu_2O$ shell for $Au@Cu_2O$ enabled the formation of a smaller void size for $Au@Cu_7S_4$. Note that Au served as the core for $Cu_2O$ deposition. Under a fixed $CuSO_4$ concentration, the employment of more Au would result in less $Cu_2O$ deposition on each Au, leading to a decrease in the $Cu_2O$ shell thickness for $Au@Cu_2O$. Three amounts of Au colloids (1.0, 3.0 or 5.0 mL) were used to prepare for $Au@Cu_2O$ with decreasing shell thicknesses. Upon the sulfidation treatment, $Au@Cu_7S_4$ with decreasing void sizes could then be obtained. The resultant $Au@Cu_7S_4$ was respectively denoted as 1-$Au@Cu_7S_4$, 3-$Au@Cu_7S_4$ and 5-$Au@Cu_7S_4$. Note that it is improbable to obtain $Au@Cu_7S_4$ with a smaller void size than that of 5-$Au@Cu_7S_4$ by using the current synthetic method. As shown in Supplementary Fig. 20, further reducing the amount of Au colloids to 7.0 mL led to the growth of 7-$Au@Cu_7S_4$ exhibiting nearly identical structural dimensions to those of 5-$Au@Cu_7S_4$. If examined closely, the individual

7-$Au@Cu_7S_4$ contained multiple Au yolks encapsulated in $Cu_7S_4$ shell, which might be due to the aggregation of Au colloids under the relatively high concentration situation. Since the yolk@shell structural integrity of 7-$Au@Cu_7S_4$ can no longer be maintained, 7-$Au@Cu_7S_4$ was not adopted for further performance comparison. For comparison purposes, pure $Cu_2O$ nanocrystals were also synthesized by using the same chemical reduction method without the employment of Au colloids. The thus-obtained pure, solid $Cu_2O$ was subjected to the same sulfidation treatment to obtain pure, hollow $Cu_7S_4$. To obtain Au+$Cu_7S_4$, Au colloids and pure $Cu_7S_4$ suspension were simply mixed, with their concentrations deliberately adjusted to equate them to those of 5-$Au@Cu_7S_4$.

### Fundamental characterizations
The microstructural features of the samples were investigated by HRTEM (JEOL, JEM-F200) equipped with SAED and EDS. The crystallographic structure was studied with XRD (Bruker, D2 Phaser). The optical properties were analyzed with a UV-visible-NIR absorption spectrophotometer (Hitachi, U-3900H) and a fluorescence spectrophotometer (Hitachi, F-4500). To measure the steady-state PL spectra, samples in a given amount were dispersed in deionized water. To observe the direct influence of Au on the PL of $Cu_7S_4$, the concentration of the $Cu_7S_4$ component for pure $Cu_7S_4$ and the three $Au@Cu_7S_4$ was adjusted to a fixed value by normalizing their absorbance at 450 nm. The chemical states of the samples were examined with X-ray photoelectron spectroscopy (XPS, Thermo Fisher Scientific, ESCALAB Xi$^+$) using a monochromatic Al $K\alpha$ X-ray source. The recorded binding energies were calibrated based on the C $1s$ peak at 284.8 eV from adventitious carbon. The band structure of the samples was determined by measuring the UPS spectra (Thermo Fisher Scientific, ESCALAB Xi$^+$) using He I (hυ = 21.22 eV) as the excitation source. The specific surface areas of the samples were measured from the $N_2$ adsorption-desorption isotherms by using the BET method (Micromeritics, ASAP2020).

### In-situ XAS measurements
XAS measurements were performed at beamline 17 C and 20 A of the Taiwan Light Source of the National Synchrotron Radiation Research Center, Taiwan. Soft XAS in total electron-yield (TEY) mode was utilized to conduct measurements under dark and irradiation conditions. Soft X-rays are characterized by a shallow penetration depth, making them suitable for collecting data that accurately represent the electronic structure in the vicinity at the Au/$Cu_7S_4$ interface of $Au@Cu_7S_4$. This region is of particular interest since it is where charge transfer and separation take place. The soft XAS data were obtained using a high-resolution spectrometer with a resolving power of approximately $8\times10^3$ in an ultrahigh-vacuum chamber with a pressure of $10^{-9}$ Torr. To enable in-situ observation of the evolution of the UDOS with light irradiation, an additional irradiation produced from a solar simulator was introduced into the chamber.

### TAS measurements
Femtosecond TAS measurements were performed on a CDP-ExciPro pump-probe transient absorption spectrometer system[44]. Briefly, a femtosecond Ti-sapphire amplified laser (Coherent Legend, USP, 795 nm, 1 kHz, 3 mJ, 35 fs) was equally split between two optical parametric amplifiers (TOPAS-C) to generate pump and probe pulses. The 1300 nm signal output of TOPAS1 was attenuated and focused on a sapphire plate to generate a broadband white light continuum, whose spectrum was limited between 550 and 950 nm using a short-pass filter. TOPAS2 was used to generate pump pulses at two different wavelengths of 500 and 1400 nm. The pump pulses were delayed with respect to the probe pulses using an optical delay stage to obtain time-dependent changes in the absorption spectra of the samples. The energies of the pump pulses at 500 and

1400 nm were set at 5.0 and 0.7 mJ cm$^{-2}$, respectively. The dispersion of the white light spectrum was corrected by measuring the optical Kerr signals of the substrates following the procedures described below. For obtaining Kerr signals, the polarization of the pump was set to 45 ° with respect to probe and an analyzer was placed in the probe path behind the substrate to detect the pump-induced changes in polarization of the transmitted probe pulses through the analyzer. Here, the observed change of polarization state of probe was a result of pump-induced anisotropy in the refractive index of the substrate. The nanosecond NIR TAS profiles of the samples were obtained by using a nanosecond transient infrared spectrometer setup[45]. Briefly, 532 and 1064 nm nanosecond pulses from a Nd:YAG laser were used as excitation sources and filtered output from a tungsten halogen lamp was used as an NIR probe. The NIR transient spectra were acquired in single channel mode using a high-speed liquid N$_2$-cooled InSb detector by scanning the grating. The decay traces at each wavelength were measured using a Ni-Scope, National Instruments (8 bit, 100 MHz detector, 30,000 points).

## Photocatalytic hydrogen production

The activity for photocatalytic hydrogen production of the samples was estimated under AM 1.5 G irradiation (100 mW cm$^{-2}$) produced by a solar simulator (Newport, LCS-100, 94011 A). Six relevant samples, including pure Au, pure Cu$_7$S$_4$, the three Au@Cu$_7$S$_4$ and Au +Cu$_7$S$_4$, were tested and compared. A given amount of the sample powder was dispersed in an aqueous electrolyte (40.0 mL) containing 5.0 vol.% methanol and 15.0 wt.% glucose in a customized quartz vessel (75 mL capacity). Before irradiation, the reaction solution was purged with Ar for 1 h. At a given time interval under light irradiation, 1.0 mL of the gas in the headspace of the vessel was collected and analyzed with gas chromatography (Bruker, SCION 436GC). Comparative experiments by placing the reaction vessel on a cold plate (AS ONE, CPS-30) were also performed to investigate the influence of the temperature control of electrolyte on the photocatalytic performance of Au@Cu$_7$S$_4$. The AQY of hydrogen production was determined by conducting experiments under monochromatic illumination produced by a 150 W xenon lamp coupled with a monochromator (Horiba, Tunable PowerArc, 0.2 m, 1200 gr mm$^{-1}$, dispersion = 5 nm). The number of incident photons was measured with an optical power meter (Newport, 843-R). The AQY was calculated by Eq. (1):

$$\text{AQY}(\%) = \frac{\text{number of reacted electrons}}{\text{number of incident photons}} \times 100\%$$
$$= \frac{2 \times \text{number of evolved hydrogen molecules}}{\text{number of incident photons}} \times 100\% \tag{1}$$

## RhB release experiments

The mass transport kinetics across the Cu$_7$S$_4$ shell were estimated by monitoring the temporal release profiles of RhB adsorbed on the samples. In a typical procedure, a given amount of the sample powder was dispersed in a RhB aqueous solution (5.0 mL, 1.0 × 10$^{-5}$ M) for 24 h to enable complete adsorption of RhB on the sample surface. The suspension was collected by centrifugation in order to remove the non-adsorbed RhB. After that, the collected precipitate was redispersed in deionized water to release RhB. The concentration of released RhB was determined by measuring the absorption spectrum of the supernatant. Note that the release rate of RhB, rather than the released amount of RhB, was measured in order to reduce the influence of surface area. Because the void size was the only variable in this comparison, the observed difference in release rate can be solely assigned to the change in the void size. The D across the Cu$_7$S$_4$ shell

can be further calculated by fitting the data of Supplementary Fig. 4 to Eq. (2):

$$D = \left(\pi R^2 / 36\right) \times \left(V_f^2 / t\right) \tag{2}$$

where R was the inner radius of Au@Cu$_7$S$_4$ and V$_f$ was the volume fraction of RhB released at time t[34,35].

## Photothermal experiments

The photothermal experiments were performed by irradiating 0.2 mL of 5-Au@Cu$_7$S$_4$-contained electrolyte with high-power lasers under various excitation wavelengths from visible to NIR region. The concentration of 5-Au@Cu$_7$S$_4$ and the composition of electrolyte followed the experimental conditions used in the photocatalytic reaction. In a typical procedure, 0.2 mL of 5-Au@Cu$_7$S$_4$-contained electrolyte was added into a vial with a capacity of 0.4 mL. The vial was deposited into a 96-well transparent plate, followed by laser irradiation at 532 nm (2.0 W cm$^{-2}$), 650 nm (1.0 W cm$^{-2}$), 785 nm (4.0 W cm$^{-2}$), 808 nm (2.0 W cm$^{-2}$), and 1064 nm (0.33 W cm$^{-2}$). The power of irradiation was adjusted to the maximal capacity of the laser in order to highlight the photothermal effect. A datalogging K/J thermometer (TES-1307) with K-type thermal couple in an accuracy of ± 0.1 °C was installed in the vial to measure the electrolyte temperature. The corresponding thermo-graph images were taken with an infrared camera (Fluke, Ti32).

## DFT simulations

The Vienna Ab initio Simulation Package (VASP)[46,47] was used to perform first-principles calculations with the projector augmented wave (PAW)[48] method based on DFT. The exchange-correlation interaction for the structure optimization and the electronic properties were described by the Perdew–Burke–Ernzerhof (PBE) function within the generalized gradient approximation (GGA)[49,50]. A cut-off kinetic energy of 420 eV was set as the plane-wave basis for the bulk Cu$_7$S$_4$ and Au primitive cells. The Brillouin zone was sampled with 6 × 6 × 5 and 18 × 18 × 18 Monkhorst-Pack k-point grids[51] for the bulk Cu$_7$S$_4$ and Au primitive cells, respectively. Furthermore, the geometric structures were relaxed until the 1.0 × 10$^{-5}$ eV electronic convergence criterion was reached. To create the structural models for DFT calculations, the crystal structure of Au and Cu$_7$S$_4$ was adopted from the Materials Project (MP) database (MP database ID of mp-81 for Au; MP database ID of mp-624299 for Cu$_7$S$_4$). In order to determine the favorable adsorption site of hydrogen, the adsorption energy (E$_{ads}$) was calculated by Eq. (3):

$$E_{ads} = E_{sys} - E^* - E_{surf} \tag{3}$$

where E$_{sys}$ and E$_{surf}$ are the total energies of the slab model after and before surface adsorption, respectively, and E* is the chemical potential of the adsorbate. The calculation of the free energy of hydrogen adsorption follows Eq. (4)[52,53]:

$$\Delta G_{H^*} = E_{ads} + \Delta E_{ZPE} - T\Delta S_{H^*} \tag{4}$$

where $\Delta E_{ZPE}$ and $\Delta S_{H^*}$ are the difference in the zero-point energy and vibrational entropy between the adsorbed hydrogen and the hydrogen in the gas phase, respectively. The calculated specific E$_{ads}$ and $\Delta E_{ZPE}$ corresponding to different adsorption sites are summarized in Supplementary Table 5. Compared to the highly mobile hydrogen in the gas phase, the adsorbed hydrogen is much immobile because it is bound to the surficial atoms. The vibration entropy of the adsorbed hydrogen is thus considered small relative to the value of the gaseous hydrogen[36,54]. This consideration gives $\Delta S_{H^*} = - 0.5 S^0_{H2}$, where $S^0_{H2}$ is the entropy of the hydrogen molecule in the gas phase under standard conditions ($S^0_{H2} = 130.68$ J mol$^{-1}$ K$^{-1}$ at 300 K and 1 bar[55]). By using the value of $S^0_{H2}$ at 300 K, the $-T\Delta S_{H^*}$ term can be further

computed as follows: $-T\Delta S_{H^*} = T \times 0.5\ S^0_{H_2} = 300$ (K) $\times 0.5 \times 130.68$ (J mol$^{-1}$ K$^{-1}$) $= 19602$ (J mol$^{-1}$) $= 19602$ (J mol$^{-1}$) $\div (1.602 \times 10^{-19}$ (J eV$^{-1}$)) $\div$ $(6.02 \times 10^{23}$ (mol$^{-1}$)) $= 0.20$ (eV). The expression of $\Delta G_{H^*}$ then follows Eq. (5):

$$\Delta G_{H^*} = E_{ads} + \Delta E_{ZPE} + 0.20 \qquad (5)$$

For the free energy calculations, a $7 \times 3\sqrt{3}$ Au(111) slab with three atomic layers and a $2 \times 2$ Cu$_7$S$_4$(010) slab with four atomic layers were used to perform energy minimization. For Au(111) slab, one axis of the primitive cell was rotated by 30 degrees to form a rectangular $1 \times \sqrt{3}$ unit cell in order to match the rectangular base of Cu$_7$S$_4$(010). To construct the structural model of Au(111) slab, this rectangular unit cell was further extended to give a $7 \times 3\sqrt{3}$ super-cell. The coordinates of the atoms constituting the structural models of Au(111), Cu$_7$S$_4$(010) and Au(111)@Cu$_7$S$_4$(010) were displayed in Supplementary Table 6. To minimize the interference of periodic slabs, a vacuum region was introduced in the supercell along the direction normal to the surface[56]. The optimized thickness of vacuum region was set as 15.54 Å for all the structural models[57]. The lattice mismatches were 3.59 and 4.86%, respectively. As displayed in Supplementary Fig. 18, various surfacial sites on pure Au, pure Cu$_7$S$_4$ and Au@Cu$_7$S$_4$ surfaces were considered to obtain the optimal hydrogen adsorption model. Note that the structural models used for DFT calculations did not completely equate to the microstructural features of the samples in terms of the structural dimensions. For DFT calculations, only a few atomic layers can be considered for modelling in order to receive convergence results. Nevertheless, the computed data can deliver atomic-scale insights into the thermodynamics and kinetics properties at the surfacial region. This information is particularly relevant to the fundamental understanding of catalytic mechanism since catalytic reactions mostly occur at the localized surface of catalysts[38–40]. On the other hand, the potential distortion of the surface slab of Au@Cu$_7$S$_4$ upon H* adsorption has also been evaluated by computing the shear strain and volumetric strain in the localized region where H* was adsorbed[58]. As displayed in Supplementary Fig. 21, the surfacial S site bonded with H* experienced appreciable lattice distortion, whereas the Cu site bonded with H* did not exhibit observable lattice distortion. The corresponding bond length was 1.36 Å for S-H* and 1.70 Å for Cu-H*. Apart from the S-H* site, the surface slab closely resembled pristine Cu$_7$S$_4$ without H* adsorption. Because S atoms exhibited a wide range of periodicity on the surface slab, this localized distortion caused by H* adsorption did not pose a significant impact on the entity of the surface slab.

## Data availability

The data that support the finding of this study are available within the paper and its Supplementary information. Source data for Figs. 2a, b, 3a–f, 4, and Supplementary Figs. 1a–c, 2, 3, 4, 5c–e, 15, 17, 19, and 22 are provided with the paper. Source data are provided with this paper.

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

## Acknowledgements

This work was financially supported by the National Science and Technology Council, Taiwan, under grants NSTC 111-2113-M-A49-038 and NSTC 112-2113-M-A49-024 (Y.H.). This work was also supported by the Center for Emergent Functional Matter Science of National Yang Ming Chiao Tung University from The Featured Areas Research Center Program within the framework of the Higher Education Sprout Project by the Ministry of Education (MOE) in Taiwan. Additional support was provided by the Tokyo Institute of Technology through the Research Center for Biomedical Engineering and the World Research Hub (WRH) Program of the International Research Frontiers Initiative at Institute of Innovative Research. This study was also supported by JSPS KAKENHI grant numbers JP23K04369 (C.C.) and JP21K04827 (T.M.C.).

## Author contributions

C.T. prepared for the samples, performed all the experiments, analyzed all the data and drafted the manuscript. S.N. carried out the TAS measurements. Y.-C.L. carried out the XAS measurements. J.K., Y.L., J.C. and S.O. carried out the DFT calculations. Y.C., Z.H. and C.H. carried out the photothermal measurements. W.H., C.L. carried out the optical measurements. C.C., M.S. and M.H.H. commented on the experimental data. T.M.C., Y.-G.L., E.W.D. and Y.H. designed the experiments and finalized the manuscript. Y.H. conceived the project and supervised the research. All authors discussed the results of the manuscript.

## Competing interests

The authors declare no competing interests.

## Additional information

[1]Department of Materials Science and Engineering, National Yang Ming Chiao Tung University, Hsinchu 300093, Taiwan. [2]Department of Applied Chemistry and Institute of Molecular Science, National Yang Ming Chiao Tung University, Hsinchu 300093, Taiwan. [3]National Synchrotron Radiation Research Center, Hsinchu 30076, Taiwan. [4]Institute of Innovative Research, Tokyo Institute of Technology, Kanagawa 226-8503, Japan. [5]Department of Photonics, National Cheng Kung University, Tainan 70101, Taiwan. [6]Department of Electrophysics, National Yang Ming Chiao Tung University, Hsinchu 300093, Taiwan. [7]Institute of Physics, National Yang Ming Chiao Tung University, Hsinchu 300093, Taiwan. [8]Department of Physics, National Changhua University of Education, Changhua 50007, Taiwan. [9]Department of Mechanical Science and Bioengineering, Osaka University, Toyonaka 560-8531, Japan. [10]Department of Chemistry, National Tsing Hua University, Hsinchu 30013, Taiwan. [11]Center for Emergent Functional Matter Science, National Yang Ming Chiao Tung University, Hsinchu 300093, Taiwan. [12]International Research Frontiers Initiative, Institute of Innovative Research, Tokyo Institute of Technology, Kanagawa 226-8503, Japan. ✉e-mail: chang.m.aa@m.titech.ac.jp; yclo@NYCU.edu.tw; lin.yg@nsrrc.org.tw; diau@nycu.edu.tw; yhsu@nycu.edu.tw; yhsu@cc.nctu.edu.tw

