## [Peer review file · Nature Communications]

Reviewers' comments:

Reviewer #1 (Remarks to the Author):

In this work, Tsao and collaborators performed a combined experimental and theoretical study on Au@Cu₇S₄ yolk@shell nanocrystals as potential photocatalysts for hydrogen production across the visible to NIR spectral region. Even though the content of this manuscript might be of potential interest for Nature Communication, there are several weaknesses that make the manuscript not publishable in such a high impact journal in this current form. The calculations presented in the supporting information are very elusive, and there is no explanation at all of the DFT results. Moreover, several drawbacks are also present in the description of the computational protocol. Based on DFT calculations, in the manuscript the authors state that the "surficial S site of Au@Cu₇S₄ held the greatest promise for proceeding with the hydrogen evolution reaction because the computed free energy of hydrogen adsorption ($\Delta G_H^* = 0.23$ eV) was closest to zero." However, neither in the supporting information nor in the manuscript, a detailed discussion of the DFT results is presented. For example, there is no information about structural parameters, such as the bond lengths of the adsorbed hydrogen to the surface sites, the potential distortion of the surface slab upon H adsorption, and how those changes influence the different adsorption energies and ΔG s.

A description of the crystal structure of both Cu₇S₄ and Au should also be given to underline the type of coordination and bond that each atom has. Based on those descriptions, the authors should motivate their choice of considering the reported adsorption sites. In this regard, I strongly recommend to rebuild figure 8 of the SI since, in its current form, it is quite difficult to visualize the structures and the corresponding adsorption sites.

The authors do not report the vacuum layer and the number of layers used for modeling the Au(111)@Cu₇S₄ slab considered in this study. Moreover, the supercell dimensions are not given. This information is crucial and cannot be missed.

Experimental parameters of the cell parameters are missing. The authors should add some references that support they optimized parameters.

The authors should also include some reference that justify their choice of considering the entropy of the adsorbed H atom small and equal to half of that of molecular hydrogen, and explain in more details the 0.20 eV value they give in the equation: $\Delta G_H^* = E_{ads} + \Delta E_{ZPE} + 0.20$ eV. Equations should be numbered.

What does 3R3 in $7 \times 3R3$ Au(111) slab mean?

The authors should add the coordinates of all the structures discussed in the manuscript.

In the manuscript, the authors state that "This feature further suggested that Cu₇S₄ could be the active site for hydrogen production on Au@Cu₇S₄.". However, the feasibility of Au@Cu₇S₄ for Hydrogen production does not depend only on the adsorption strength of H atoms, but also on the energy barriers involved in the hydrogen production and release. The evaluation of such barriers relies on the calculation of the corresponding transition state. The authors should mention this aspect. For the sake of simplicity, I would also suggest to add the ΔG energies to Table 5 of the SI.

Reviewer #2 (Remarks to the Author):

This manuscript describes dual-plasmonic Au@Cu₇S₄ yolk@shell nanocrystals for photocatalytic hydrogen production across visible to near infrared spectral region. Authors attribute the facilitated hydrogen production of dual-plasmonic Au@Cu₇S₄ yolk@shell to distinctive absorption in the visible and NIR regions, the advantageous features of yolk@shell nanostructures and the prevalence of long-lived charge separation states prolongs the lifetime of the delocalized electrons. To justify the charge transfer dynamics, a series of characterizations have been performed. However, several aspects related to the photocatalytic experiments and other linked issues are unclear.

I do not recommend the acceptance of this article in Nature communications. In order to justify this decision, I provide to the authors some comments that should be taking into account for its further

publication.

1. The particle size statistics of all sample void and Au particles should be given to exclude their influence on the performance.
2. Only the post-reaction data of SEM images and XPS spectra cannot prove that the catalyst has considerably high stability, so it is necessary to give the HRTEM, STEM-HAADF, EDS mapping and the particle size statistics of the shell (Cu₇S₄) and the yolk (Au particles).
3. There is lack of superiority in the performance of sulphide materials compared to those recent publications, such as Nat Commun 13, 1287 (2022) ; Nat Commun 12, 1343 (2021) ; Nat Commun 13, 4600 (2022) ; Adv. Funct. Mater. 2019, 1901958 ; Adv. Energy Mater. 2016, 1600464 ; Adv. Mater. 2022, 34, 2108475.
4. Is the light intensity of different wavelengths consistent during the in-situ synchrotron radiation test? Besides, the intensity of the corresponding L-edge X-ray absorption spectra of Au and Cu reflects the oxidation state of the corresponding elements. The obtained different intensities of absorption spectra irradiated with different wavelengths of light can only explain the intensity of electron gain and loss of corresponding elements, and it cannot be concluded that "there are three different charge transfer scenarios". Have the authors performed controlled experiments in which the order of different wavelengths of light is varied? The authors mentioned on page 21 that the LSPR wavelength of Au in 5-Au@Cu₇S₄ is between 522.2-755.8 nm, which just corresponds to the visible light region and the infrared region, which means that the light absorption is the strongest in this wavelength range (from Supplementary Fig. 1 can also be concluded that this range has the strongest light absorption), so the synchrotron radiation spectrum shows a large change when irradiated with visible light and infrared light.
5. The 5-Au@Cu₇S₄ with the smallest void size has the best performance. If it is further reduced, will the performance be better?
6. More information on the structure and chemical state of the material should be provided.
7. In Supplementary Fig. 1c on page 21, the author mentioned that XPS proved that the valence state of Cu in the material has always maintained a +2 valence. However, the authors did not give the XPS spectrum of the sample before the reaction. Secondly, the valence state of Cu is difficult to pass through XPS to distinguish +2 and +1 valences, thus it is necessary to provide Auger electron spectra before and after the reaction.
8. The determination of the active site is only through theoretical calculations. Is there any corresponding experimental evidence? Besides, the established theoretical model does not correspond to the experiment.
9. Through a series of experiments, the authors prove that the material has excellent photogenerated carrier separation and inhibited recombination capabilities in the infrared and visible light regions. It is not optimal under AM1.5, why is the performance test under AM1.5? Secondly, the authors did not rule out the thermal effect during the test.
10. Please supplement electrochemical impedance spectroscopy, photocurrent spectroscopy, and steady-state fluorescence spectroscopy to illustrate the optoelectronic properties of materials.

Reviewer #3 (Remarks to the Author):

The authors reported that the Au@Cu₇S₄ yolk@shell nanostructure as a dual-plasmonic photocatalyst to achieve remarkable hydrogen production under visible and NIR illumination, including the synthesis, microstructural, LSPR property and charge transfer dynamics of the nanostructures. The results indicated that the Au@Cu₇S₄ exhibits superior hydrogen production. I think that the results of this manuscript should be interesting to the readership of Nature Communications. However, the following questions should be properly addressed by the authors before this manuscript can be accepted for publication:

1. The author claims that the Au@Cu₇S₄ nanostructures comprised a movable Au particle surrounded by a hollow Cu₇S₄. As Cu₇S₄ and Au are brought in contact, whether the band bending is related to the chemical bonding behavior between Au particles and Cu₇S₄, and is interrelated Au particles are in inside or outside of Cu₇S₄ shell.

2. Hot electron-hole pairs generated by LSPR have an important impact for photocatalytic hydrogen production, please briefly describe the effect of Au/Cu₇S₄ interface and the possible mechanism.
3. Au colloids and pure Cu₇S₄ suspension was simply mixed and denoted as Au+Cu₇S₄, as a comparison, Au+Cu₇S₄ and Au@Cu₇S₄ yolk@shell nanocrystals are incomparable since their density is different.

Summary of changes and replies to the Reviewers' comments

Reviewer #1

Comments: In this work, Tsao and collaborators performed a combined experimental and theoretical study on Au@Cu₇S₄ yolk@shell nanocrystals as potential photocatalysts for hydrogen production across the visible to NIR spectral region. Even though the content of this manuscript might be of potential interest for Nature Communication, there are several weaknesses that make the manuscript not publishable in such a high impact journal in this current form. The calculations presented in the supporting information are very elusive, and there is no explanation at all of the DFT results. Moreover, several drawbacks are also present in the description of the computational protocol.

Author Response: We thank the Reviewer for giving the critical yet constructive comments on the results of DFT calculations. Here, we would like to highlight that the main scope of the current work is experimentally developing a dual plasmonic photocatalyst paradigm that can be responsive to visible and NIR irradiation for realizing solar hydrogen production. This work is not aimed to create a theoretical approach to the computational study of the photocatalytic mechanism. There were totally 13 figures present in the original manuscript, while only 2 supplementary figures were associated with DFT calculations. It was by all means obvious that the DFT calculations were carried out to merely provide supporting data to consolidate the experimental findings. For sure the theoretical considerations from DFT calculations can be as important as the deductive interpretations from experimental findings. With all due respect, but we cannot assent to the Reviewer's biased act on neglecting all of the experimental findings yet criticizing only the very minor part of DFT calculations. Nevertheless, we have provided additional computational data and interpretations in the revised manuscript in order to fully address the Reviewer's comments on DFT calculations. We sincerely hope the Reviewer could more appreciate the significance and impact of the experimental findings that the current work can bring forth to the community.

Comment 1: Based on DFT calculations, in the manuscript the authors state that the "surficial S site of Au@Cu₇S₄ held the greatest promise for proceeding with the hydrogen evolution reaction because the computed free energy of hydrogen adsorption ($\Delta G_{H^*} = 0.23$ eV) was closest to zero." However, neither in the supporting information nor in the manuscript, a detailed discussion of the DFT results is presented. For example, there is no information about structural parameters, such as the bond lengths of the adsorbed hydrogen to the surface sites, the potential distortion

of the surface slab upon H adsorption, and how those changes influence the different adsorption energies and ΔG s.

Response 1: We thank the Reviewer for bringing up this important question. We have provided more detailed discussions along with additional computational data to interpret the results of DFT calculations. In this work, DFT calculations were performed to fundamentally understand the origin of the superiority of Au@Cu₇S₄ in hydrogen production. Since the real crystals of Au@Cu₇S₄ are large and highly complex, the structural models used for DFT calculations were simplified to obtain the convergence results. The optimized structures of pure Au, pure Cu₇S₄ and Au@Cu₇S₄ were displayed in Supplementary Fig. 10, and the calculation details and results were shown in the Supplementary Information and Supplementary Fig. 11. Here, the Gibbs free energy of hydrogen adsorption (ΔG_{H^*}) on the three structural models was compared to look into the overall thermodynamics behaviors of hydrogen production. In general, the ideal catalysts suited for hydrogen production should have a $|\Delta G_{H^*}|$ value close to zero.¹ As summarized in Supplementary Fig. 11 and Table 5, the computed ΔG_{H^*} value for Cu and S site of pure Cu₇S₄ was -5.17 and -5.05 eV, respectively, indicating that hydrogen was easy to adsorb on Cu₇S₄ to form strong Cu₇S₄-H* bonding. Since the Cu₇S₄-H* bonds were considerably stable, the subsequent desorption of H* would be suppressed to hinder hydrogen evolution.² In contrast, pure Au possessed an appropriate ΔG_{H^*} for the adsorption and desorption of H*; the relatively small ΔG_{H^*} (0.46 eV) suggested its promise as an intriguing catalyst for hydrogen production. The inadequacy for responding to light irradiation, however, limited the practice of pure Au in solar hydrogen production. Compared to pure Cu₇S₄, Au@Cu₇S₄ held a much lower $|\Delta G_{H^*}|$ value. The computed ΔG_{H^*} was 0.80 eV for Cu site and 0.23 eV for S site, signifying that both Cu and S sites of Au@Cu₇S₄ were thermodynamically favorable for proceeding with hydrogen evolution reaction. For Au@Cu₇S₄, the electronic interactions between Au and Cu₇S₄ may optimize hydrogen evolution by mediating hydrogen adsorption and desorption on Cu₇S₄.^{2,3,4,5} This feature further suggested that Cu₇S₄ could be the active site for solar hydrogen production on Au@Cu₇S₄.

On the other hand, the potential distortion of the surface slab of Au@Cu₇S₄ upon H* adsorption has also been evaluated by computing the shear strain and volumetric strain in the localized region where H* was adsorbed.⁶ As displayed in Supplementary Fig. 12, the surficial S site bonded with H* experienced appreciable lattice distortion, whereas the Cu site bonded with H* did not exhibit observable lattice distortion. The corresponding bond length was 1.36 Å for S-H* and 1.70 Å for Cu-H*. Apart from the S-H* site, the surface slab closely resembled pristine Cu₇S₄ without H* adsorption. Because S atoms exhibited a wide range of periodicity on the surface slab, this

localized distortion caused by H* adsorption did not pose a significant impact on the entity of surface slab. We have included the above discussions and Supplementary Fig. 12 in the revised manuscript.

Supplementary Fig. 12 Distortion of the surface slab of Au@Cu₇S₄ upon H* adsorption. **a** Computed shear strain, **b** computed volumetric strain in the localized region where H* was adsorbed on S site. **d** Computed shear strain, **e** computed volumetric strain in the localized region where H* was adsorbed on Cu site. **c** and **f** show the atomic arrangement for S-H* and Cu-H* case, respectively.

Comment 2: A description of the crystal structure of both Cu₇S₄ and Au should also be given to underline the type of coordination and bond that each atom has. Based on those descriptions, the authors should motivate their choice of considering the reported adsorption sites. In this regard, I strongly recommend to rebuild figure 8 of the SI since, in its current form, it is quite difficult to visualize the structures and the corresponding adsorption sites.

Response 2: We thank the Reviewer for pointing out these deficiencies. To create the structural models for DFT calculations, we adopted the crystal structure of Au and Cu₇S₄ from the Materials Project (MP) database (MP database ID of mp-81 for Au; MP database ID of mp-624299 for Cu₇S₄). This information has been included in the revised manuscript. We have also revised Supplementary Fig. 10 as follows by labeling the visualization directions for various surfacial sites on the supercells.

Supplementary Fig. 10 Structural models used for DFT calculations. Top and side views for various surfacial sites: **a** Au(111), **b** Cu₇S₄(010)-Cu site, **c** Cu₇S₄(010)-S site, **d** Au(111)@Cu₇S₄(010)-Cu site, and **e** Au(111)@Cu₇S₄(010)-S site. For Au(111), top and side views are directed along [111] and $[2\bar{1}\bar{1}]$, respectively. For Cu₇S₄(010), top and side views are directed along [010] and [100], respectively. The optimal adsorption site of hydrogen atom for each surface models is marked by gray arrow and red circle. The blue, yellow, gold, and red spheres represent the Cu, S, Au and H atoms, respectively.

Comment 3: The authors do not report the vacuum layer and the number of layers used for modeling the Au(111)@Cu₇S₄ slab considered in this study. Moreover, the supercell dimensions are not given. This information is crucial and cannot be missed.

Response 3: We thank the Reviewer for the attentive examination on this point. We have revised the experimental descriptions of DFT calculations in the revised manuscript as follows. For the free energy calculations, a $7 \times 3\sqrt{3}$ Au(111) slab with three atomic layers and a 2×2 Cu₇S₄(010) slab with four atomic layers were used to perform energy minimization. To minimize the interference of periodic slabs, a vacuum region was introduced in the supercell along the direction normal to the surface.⁷ The optimized thickness of vacuum region was set as 15.54 Å for all the structural models.⁸ We have also revised Supplementary Fig. 10 by including the supercell dimensions.

Comment 4: Experimental parameters of the cell parameters are missing. The authors should add some references that support they optimized parameters.

Response 4: We have provided the detailed the experimental parameters used to construct the supercell. Please refer to Response 2 and Response 3 to the Reviewer #1.

Comment 5: The authors should also include some reference that justify their choice of considering the entropy of the adsorbed H atom small and equal to half of that of molecular hydrogen, and explain in more details the 0.20 eV value they give in the equation: $\Delta G_{H^*} = E_{ads} + \Delta E_{ZPE} + 0.20 \text{ eV}$.

Response 5: We thank the Reviewer for the constructive comments. We have added two new references to justify the simplified term of the vibrational entropy between the adsorbed hydrogen and the hydrogen in the gas phase. Compared to the highly mobile hydrogen in the gas phase, the adsorbed hydrogen is much immobile because it is bound to the surficial atoms. The vibration entropy of the adsorbed hydrogen is thus considered small relative to the value of the gaseous hydrogen.^{1,9} This consideration gives $\Delta S_{H^*} = -0.5 S^0_{H_2}$, where $S^0_{H_2}$ is the entropy of the hydrogen molecule in the gas phase under standard conditions ($S^0_{H_2} = 130.68 \text{ J mol}^{-1} \text{ K}^{-1}$ at 300 K and 1 bar¹⁰). By using the value of $S^0_{H_2}$ at 300 K, the $-T\Delta S_{H^*}$ term can be further computed as follows: $-T\Delta S_{H^*} = T \times 0.5 S^0_{H_2} = 300 \text{ (K)} \times 0.5 \times 130.68 \text{ (J mol}^{-1} \text{ K}^{-1}) = 19602 \text{ (J mol}^{-1}) = 19602 \text{ (J mol}^{-1}) \div (1.602 \times 10^{-19} \text{ (J eV}^{-1})) \div (6.02 \times 10^{23} \text{ (mol}^{-1})) = 0.20 \text{ (eV)}$. The expression of ΔG_{H^*} then follows $\Delta G_{H^*} = E_{ads} + \Delta E_{ZPE} + 0.20 \text{ eV}$. We have added the above descriptions in the revised manuscript.

Comment 6: Equations should be numbered.

Response 6: We have numbered all the equations noted in the manuscript.

Comment 7: What does 3R3 in $7 \times 3R3$ Au(111) slab mean?

Response 7: For Au(111) slab, one axis of the primitive cell was rotated by 30 degrees to form a rectangular $1 \times \sqrt{3}$ unit cell in order to match the rectangular base of Cu₇S₄(010). To construct the structural model of Au(111) slab, this rectangular unit cell was further extended to give a $7 \times 3\sqrt{3}$ supercell. The “3R3” was meant to represent " $3\sqrt{3}$ ". We have modified the notation from $7 \times 3R3$ to $7 \times 3\sqrt{3}$ to avoid confusion.

Comment 8: The authors should add the coordinates of all the structures discussed in the manuscript.

Response 8: The coordinates of the atoms constituting the structural models of Au(111), Cu₇S₄(010) and Au(111)@Cu₇S₄(010) have been displayed as Supplementary Table 6 in the revised manuscript.

Supplementary Table 6. Coordinates of the atoms constituting the structural models of Au(111), Cu₇S₄(010) and Au(111)@Cu₇S₄(010).

Au(111)	Coordinates (Å)		
Atoms	x	y	z
Au	0	0	1.96697
Au	0	5.088447	1.96697
Au	0	10.1769	1.96697
Au	2.937816	0	1.96697
Au	2.937816	5.088447	1.96697
Au	2.937816	10.1769	1.96697
Au	5.875633	0	1.96697
Au	5.875633	5.088447	1.96697
Au	5.875633	10.1769	1.96697
Au	8.813449	0	1.96697
Au	8.813449	5.088447	1.96697
Au	8.813449	10.1769	1.96697
Au	11.75127	0	1.96697
Au	11.75127	5.088447	1.96697
Au	11.75127	10.1769	1.96697
Au	14.68908	0	1.96697
Au	14.68908	5.088447	1.96697
Au	14.68908	10.1769	1.96697
Au	17.6269	0	1.96697
Au	17.6269	5.088447	1.96697
Au	17.6269	10.1769	1.96697
Au	1.468908	2.544224	1.96697
Au	1.468908	7.632671	1.96697
Au	1.468908	12.72112	1.96697
Au	4.406725	2.544224	1.96697
Au	4.406725	7.632671	1.96697
Au	4.406725	12.72112	1.96697
Au	7.344541	2.544224	1.96697
Au	7.344541	7.632671	1.96697
Au	7.344541	12.72112	1.96697
Au	10.28236	2.544224	1.96697
Au	10.28236	7.632671	1.96697

Au	10.28236	12.72112	1.96697
Au	13.22018	2.544224	1.96697
Au	13.22018	7.632671	1.96697
Au	13.22018	12.72112	1.96697
Au	16.15799	2.544224	1.96697
Au	16.15799	7.632671	1.96697
Au	16.15799	12.72112	1.96697
Au	19.09581	2.544224	1.96697
Au	19.09581	7.632671	1.96697
Au	19.09581	12.72112	1.96697
Au	1.468908	0.84806	4.378585
Au	1.468908	5.936507	4.378585
Au	1.468908	11.02496	4.378585
Au	4.406725	0.84806	4.378585
Au	4.406725	5.936507	4.378585
Au	4.406725	11.02496	4.378585
Au	7.344541	0.84806	4.378585
Au	7.344541	5.936507	4.378585
Au	7.344541	11.02496	4.378585
Au	10.28236	0.84806	4.378585
Au	10.28236	5.936507	4.378585
Au	10.28236	11.02496	4.378585
Au	13.22018	0.84806	4.378585
Au	13.22018	5.936507	4.378585
Au	13.22018	11.02496	4.378585
Au	16.15799	0.84806	4.378585
Au	16.15799	5.936507	4.378585
Au	16.15799	11.02496	4.378585
Au	19.09581	0.84806	4.378585
Au	19.09581	5.936507	4.378585
Au	19.09581	11.02496	4.378585
Au	0	3.392284	4.378585
Au	0	8.480731	4.378585
Au	0	13.56918	4.378585
Au	2.937816	3.392284	4.378585
Au	2.937816	8.480731	4.378585
Au	2.937816	13.56918	4.378585
Au	5.875633	3.392284	4.378585

Au	5.875633	8.480731	4.378585
Au	5.875633	13.56918	4.378585
Au	8.813449	3.392284	4.378585
Au	8.813449	8.480731	4.378585
Au	8.813449	13.56918	4.378585
Au	11.75127	3.392284	4.378585
Au	11.75127	8.480731	4.378585
Au	11.75127	13.56918	4.378585
Au	14.68908	3.392284	4.378585
Au	14.68908	8.480731	4.378585
Au	14.68908	13.56918	4.378585
Au	17.6269	3.392284	4.378585
Au	17.6269	8.480731	4.378585
Au	17.6269	13.56918	4.378585
Au	0	1.696064	6.775393
Au	0	6.784511	6.790393
Au	0	11.87296	6.790393
Au	2.937816	1.696064	6.790393
Au	2.937816	6.784511	6.790393
Au	2.937816	11.87296	6.790393
Au	5.875633	1.696064	6.790393
Au	5.875633	6.784511	6.790393
Au	5.875633	11.87296	6.790393
Au	8.813449	1.696064	6.790393
Au	8.813449	6.784511	6.790393
Au	8.813449	11.87296	6.790393
Au	11.75127	1.696064	6.790393
Au	11.75127	6.784511	6.790393
Au	11.75127	11.87296	6.790393
Au	14.68908	1.696064	6.790393
Au	14.68908	6.784511	6.790393
Au	14.68908	11.87296	6.790393
Au	17.6269	1.696064	6.790393
Au	17.6269	6.784511	6.790393
Au	17.6269	11.87296	6.790393
Au	1.468908	4.240288	6.790393
Au	1.468908	9.328736	6.790393
Au	1.468908	14.41718	6.790393

Au	4.406725	4.240288	6.790393
Au	4.406725	9.328736	6.790393
Au	4.406725	14.41718	6.790393
Au	7.344541	4.240288	6.790393
Au	7.344541	9.328736	6.790393
Au	7.344541	14.41718	6.790393
Au	10.28236	4.240288	6.790393
Au	10.28236	9.328736	6.790393
Au	10.28236	14.41718	6.790393
Au	13.22018	4.240288	6.790393
Au	13.22018	9.328736	6.790393
Au	13.22018	14.41718	6.790393
Au	16.15799	4.240288	6.790393
Au	16.15799	9.328736	6.790393
Au	16.15799	14.41718	6.790393
Au	19.09581	4.240288	6.790393
Au	19.09581	9.328736	6.790393
Au	19.09581	14.41718	6.790393

Cu ₇ S ₄ (010) Coordinates (Å)			
Atoms	x	y	z
Cu	4.059319	1.782336	9.282761
Cu	4.059319	12.590119	9.282761
Cu	11.9759	1.782336	9.282761
Cu	11.9759	12.590119	9.282761
Cu	7.806314	7.111293	5.199943
Cu	7.806314	17.919077	5.199943
Cu	15.722894	7.111293	5.199943
Cu	15.722894	17.919077	5.199943
Cu	3.892753	9.335617	9.665418
Cu	3.892753	20.143399	9.665418
Cu	11.809333	9.335617	9.665418
Cu	11.809333	20.143399	9.665418
Cu	4.017622	4.82912	9.931912
Cu	4.017622	15.636903	9.931912
Cu	11.934203	4.82912	9.931912
Cu	11.934203	15.636903	9.931912

Cu	2.197308	1.763536	3.911023
Cu	2.197308	12.571319	3.911023
Cu	10.113889	1.763536	3.911023
Cu	10.113889	12.571319	3.911023
Cu	0.101029	3.621555	9.282761
Cu	0.101029	14.429338	9.282761
Cu	8.017611	3.621555	9.282761
Cu	8.017611	14.429338	9.282761
Cu	2.684867	4.746417	3.366695
Cu	2.684867	15.554199	3.366695
Cu	10.601448	4.746417	3.366695
Cu	10.601448	15.554199	3.366695
Cu	6.570375	0.677286	7.093046
Cu	6.570375	11.485068	7.093046
Cu	14.486956	0.677286	7.093046
Cu	14.486956	11.485068	7.093046
Cu	6.721984	9.085605	10.112027
Cu	6.721984	19.893388	10.112027
Cu	14.638564	9.085605	10.112027
Cu	14.638564	19.893388	10.112027
Cu	1.711452	7.225064	7.791872
Cu	1.711452	18.032846	7.791872
Cu	9.628033	7.225064	7.791872
Cu	9.628033	18.032846	7.791872
Cu	3.84835	6.353735	5.199943
Cu	3.84835	17.161517	5.199943
Cu	11.76493	6.353735	5.199943
Cu	11.76493	17.161517	5.199943
Cu	5.232105	5.9511	7.347933
Cu	5.232105	16.758883	7.347933
Cu	13.148687	5.9511	7.347933
Cu	13.148687	16.758883	7.347933
Cu	5.669743	8.98661	7.791872
Cu	5.669743	19.794392	7.791872
Cu	13.586324	8.98661	7.791872
Cu	13.586324	19.794392	7.791872
Cu	0.059332	0.574771	9.931912
Cu	0.059332	11.382554	9.931912

Cu	7.975912	0.574771	9.931912
Cu	7.975912	11.382554	9.931912
Cu	1.242119	8.990669	9.717354
Cu	1.242119	19.798453	9.717354
Cu	9.1587	8.990669	9.717354
Cu	9.1587	19.798453	9.717354
Cu	2.763693	7.126069	10.112027
Cu	2.763693	17.933851	10.112027
Cu	10.680274	7.126069	10.112027
Cu	10.680274	17.933851	10.112027
Cu	7.851043	6.876055	9.665418
Cu	7.851043	17.683837	9.665418
Cu	15.767625	6.876055	9.665418
Cu	15.767625	17.683837	9.665418
Cu	2.612086	4.726605	7.093046
Cu	2.612086	15.534388	7.093046
Cu	10.528666	4.726605	7.093046
Cu	10.528666	15.534388	7.093046
Cu	6.643157	0.657475	3.366695
Cu	6.643157	11.465257	3.366695
Cu	14.559737	0.657475	3.366695
Cu	14.559737	11.465257	3.366695
Cu	3.848024	9.10038	5.199943
Cu	3.848024	19.908162	5.199943
Cu	11.764604	9.10038	5.199943
Cu	11.764604	19.908162	5.199943
Cu	4.587287	1.866531	5.199943
Cu	4.587287	12.674312	5.199943
Cu	12.503867	1.866531	5.199943
Cu	12.503867	12.674312	5.199943
Cu	6.155598	3.640355	3.911023
Cu	6.155598	14.448138	3.911023
Cu	14.072178	3.640355	3.911023
Cu	14.072178	14.448138	3.911023
Cu	2.171797	1.770734	6.440424
Cu	2.171797	12.578517	6.440424
Cu	10.088378	1.770734	6.440424
Cu	10.088378	12.578517	6.440424

Cu	6.130088	3.633157	6.440424
Cu	6.130088	14.44094	6.440424
Cu	14.046668	3.633157	6.440424
Cu	14.046668	14.44094	6.440424
Cu	5.20041	7.221004	9.717354
Cu	5.20041	18.028786	9.717354
Cu	13.11699	7.221004	9.717354
Cu	13.11699	18.028786	9.717354
Cu	7.80664	9.857939	5.199943
Cu	7.80664	20.66572	5.199943
Cu	15.723221	9.857939	5.199943
Cu	15.723221	20.66572	5.199943
Cu	1.273815	10.260573	7.347933
Cu	1.273815	21.068354	7.347933
Cu	9.190395	10.260573	7.347933
Cu	9.190395	21.068354	7.347933
Cu	0.628997	3.537361	5.199943
Cu	0.628997	14.345143	5.199943
Cu	8.545578	3.537361	5.199943
Cu	8.545578	14.345143	5.199943
S	1.668884	5.564122	5.199943
S	1.668884	16.371905	5.199943
S	9.585465	5.564122	5.199943
S	9.585465	16.371905	5.199943
S	7.887712	2.426093	11.06357
S	7.887712	13.233875	11.06357
S	15.804294	2.426093	11.06357
S	15.804294	13.233875	11.06357
S	6.188826	5.280578	9.286794
S	6.188826	16.088359	9.286794
S	14.105407	5.280578	9.286794
S	14.105407	16.088359	9.286794
S	3.999892	2.906211	7.185694
S	3.999892	13.713994	7.185694
S	11.916473	2.906211	7.185694
S	11.916473	13.713994	7.185694
S	3.859548	7.818937	7.101452
S	3.859548	18.626719	7.101452

S	11.776129	7.818937	7.101452
S	11.776129	18.626719	7.101452
S	5.790904	0.122605	9.199638
S	5.790904	10.930388	9.199638
S	13.707485	0.122605	9.199638
S	13.707485	10.930388	9.199638
S	7.84313	8.306307	3.214252
S	7.84313	19.114089	3.214252
S	15.759711	8.306307	3.214252
S	15.759711	19.114089	3.214252
S	1.832615	5.281286	9.199638
S	1.832615	16.089068	9.199638
S	9.749195	5.281286	9.199638
S	9.749195	16.089068	9.199638
S	7.817838	8.392736	7.101452
S	7.817838	19.200517	7.101452
S	15.734419	8.392736	7.101452
S	15.734419	19.200517	7.101452
S	3.929423	2.977797	11.06357
S	3.929423	13.785581	11.06357
S	11.846002	2.977797	11.06357
S	11.846002	13.785581	11.06357
S	2.085596	10.636959	5.184943
S	2.085596	21.444742	5.199943
S	10.002177	10.636959	5.199943
S	10.002177	21.444742	5.199943
S	0.041602	2.49768	7.185694
S	0.041602	13.305462	7.185694
S	7.958183	2.49768	7.185694
S	7.958183	13.305462	7.185694
S	2.230536	0.123313	9.286794
S	2.230536	10.931095	9.286794
S	10.147117	0.123313	9.286794
S	10.147117	10.931095	9.286794
S	3.884839	7.905366	3.214252
S	3.884839	18.713148	3.214252
S	11.801421	7.905366	3.214252
S	11.801421	18.713148	3.214252

S	6.043886	5.574714	5.199943
S	6.043886	16.382495	5.199943
S	13.960467	5.574714	5.199943
S	13.960467	16.382495	5.199943
S	5.627174	10.647552	5.199943
S	5.627174	21.455334	5.199943
S	13.543756	10.647552	5.199943
S	13.543756	21.455334	5.199943

Au(111)@Cu ₇ S ₄ (010)			
Coordinates (Å)			
Atoms	x	y	z
Cu	4.551911	1.343007	17.412482
Cu	4.202546	12.331765	17.495359
Cu	12.100204	2.560924	17.407057
Cu	12.555309	13.044446	17.257728
Cu	7.675662	6.775357	13.217975
Cu	9.270526	17.868183	12.267991
Cu	0.19339	6.949001	13.549849
Cu	0.198133	16.348476	11.842364
Cu	3.614653	9.877798	16.856604
Cu	3.26075	20.499654	17.319701
Cu	11.882498	10.467762	17.036191
Cu	12.178261	21.073865	16.643374
Cu	4.672048	4.481884	17.58838
Cu	4.283963	15.507814	17.657206
Cu	12.194503	5.063643	18.622021
Cu	12.25385	15.734356	17.97793
Cu	1.603186	2.232664	11.996421
Cu	3.176063	12.730728	12.031311
Cu	8.80531	2.239793	11.667848
Cu	9.916723	12.526844	11.954639
Cu	0.253957	4.391138	17.858242
Cu	0.359259	14.728502	17.469147
Cu	8.130823	4.719572	17.79056
Cu	8.129706	15.345835	17.322395
Cu	2.841296	4.49819	12.020247
Cu	3.054861	15.233941	11.941115

Cu	11.076526	4.655251	11.921399
Cu	11.072534	14.780503	12.017731
Cu	6.491583	1.459487	15.569526
Cu	6.34362	11.761915	15.82599
Cu	14.24632	1.278392	15.822427
Cu	14.371342	11.612998	15.904122
Cu	5.571918	8.345859	17.691928
Cu	7.13158	20.695681	16.787239
Cu	13.925747	8.690172	17.74734
Cu	14.287438	19.368314	17.229321
Cu	1.825575	9.485606	14.623008
Cu	2.037722	19.225672	13.785635
Cu	9.83071	6.873014	15.419425
Cu	11.087528	17.957331	14.21873
Cu	3.723799	6.583131	13.268453
Cu	5.19899	17.758417	12.262154
Cu	12.107475	6.803788	13.101542
Cu	14.154917	18.527758	11.807041
Cu	5.839125	4.022635	15.279993
Cu	4.890498	17.144677	15.603842
Cu	14.29708	5.18803	16.151452
Cu	14.023387	16.753009	15.275744
Cu	5.879465	9.139695	15.093326
Cu	4.966658	21.077256	15.545453
Cu	13.634179	9.209822	15.256597
Cu	14.389445	19.371511	14.337807
Cu	0.008644	1.706085	17.82446
Cu	0.00032	12.077451	18.171793
Cu	8.464047	1.549608	17.3668
Cu	8.123501	12.073558	17.735103
Cu	1.016381	9.601874	17.039991
Cu	0.604362	20.832066	16.988402
Cu	7.68478	9.680687	16.822202
Cu	9.631217	20.840554	16.63321
Cu	2.865097	7.350417	16.974266
Cu	2.950791	17.889459	17.094818
Cu	9.755551	8.08795	17.776972
Cu	9.697527	18.201179	16.499007

Cu	7.493759	7.001984	16.568779
Cu	7.273882	17.924505	16.102404
Cu	0.323406	7.002745	17.22426
Cu	0.434256	17.309269	16.839969
Cu	2.461313	4.846628	16.275382
Cu	2.272324	15.73337	15.850352
Cu	10.667218	4.849303	16.656649
Cu	10.606848	15.870915	15.707068
Cu	6.204477	1.567487	11.819723
Cu	5.633685	12.440877	11.868145
Cu	14.269188	2.592138	11.851197
Cu	14.09217	11.912834	11.803953
Cu	3.934183	10.39125	12.727695
Cu	3.932093	0.238784	12.350239
Cu	11.532156	10.457844	12.009969
Cu	11.999178	21.389017	13.072449
Cu	4.084262	2.451608	10.894818
Cu	4.616044	12.699111	14.127402
Cu	11.369559	2.010606	11.910851
Cu	12.139618	12.72173	13.296076
Cu	5.63187	4.103874	12.073824
Cu	5.546235	15.069769	12.072806
Cu	13.594338	5.030848	11.821214
Cu	13.675157	14.456747	12.114275
Cu	1.600179	3.138855	14.38791
Cu	2.015387	13.76443	14.02967
Cu	9.826845	3.234239	13.846157
Cu	9.830524	12.576863	14.609117
Cu	7.307525	2.822577	13.595095
Cu	5.956393	14.488839	15.343321
Cu	13.765424	3.493939	14.173192
Cu	13.72176	13.95954	14.948845
Cu	4.99912	6.388958	15.917566
Cu	5.418234	18.891784	17.679601
Cu	12.220973	7.040989	16.671268
Cu	12.193228	18.10864	16.487447
Cu	7.817995	11.065888	12.099589
Cu	8.010206	21.481995	12.786543

Cu	15.802091	10.307227	12.714323
Cu	15.795513	0.380011	12.866807
Cu	1.944894	11.805302	15.841049
Cu	2.371473	1.094484	16.010404
Cu	9.952542	9.767896	15.435074
Cu	10.55409	1.394243	15.647093
Cu	0.240638	4.474736	12.369157
Cu	15.655613	15.192918	14.040291
Cu	8.145423	4.6071	11.982164
Cu	8.13634	15.632896	14.545758
S	1.866885	5.37303	13.904874
S	1.910255	16.107577	13.600953
S	9.691722	5.498395	13.457437
S	10.051285	16.180963	13.441241
S	15.093449	3.139158	19.445852
S	15.172439	13.924609	19.329262
S	7.445881	2.876908	18.999459
S	7.25524	13.804488	18.909691
S	6.086235	6.21186	18.080314
S	6.163657	16.813322	17.643683
S	13.931158	6.482379	18.015163
S	14.030681	17.151648	17.47955
S	3.890102	3.036793	15.989872
S	3.765084	13.849941	15.820736
S	11.687386	3.423288	15.343873
S	11.571609	13.779648	15.393167
S	3.770874	8.300633	15.164873
S	3.694107	19.168568	15.433541
S	11.650894	8.225994	14.889631
S	12.232876	19.752474	14.755711
S	5.486309	21.019876	18.212456
S	5.789599	10.56572	17.801775
S	14.207837	21.536782	17.641914
S	13.837759	10.871905	17.992939
S	7.816518	7.846239	11.220464
S	7.970214	19.49891	11.642141
S	0.199625	8.296222	11.711058
S	0.125458	19.758183	12.539853

S	1.960801	5.828798	18.280563
S	2.039499	16.183275	18.147767
S	10.000967	5.974411	18.414013
S	10.139181	16.515201	17.831745
S	8.029856	8.330685	14.886101
S	8.433711	19.617501	15.275161
S	15.792036	8.398812	15.459774
S	0.45253	19.149601	15.508293
S	5.398098	2.835366	19.012014
S	5.202526	13.807084	18.964877
S	12.988285	3.131815	19.3734
S	13.077561	14.044128	19.175973
S	1.873248	11.433246	13.443981
S	2.052706	0.929307	13.768465
S	9.690475	10.757027	13.335701
S	9.979213	0.916552	13.397291
S	15.583094	3.223428	15.903779
S	15.82209	13.507057	15.593988
S	7.949768	3.184154	15.798458
S	8.013228	13.516451	15.76211
S	1.984366	0.538569	18.165362
S	2.112125	11.220305	18.083606
S	10.575827	0.7123	17.891456
S	9.740309	10.354079	17.718405
S	3.685585	8.340328	11.854842
S	3.816243	19.494867	12.380692
S	11.719574	8.23088	11.386667
S	12.581821	20.159326	11.232557
S	5.752698	5.644467	13.703157
S	6.088286	16.361352	13.812407
S	14.027572	5.781332	13.858115
S	13.983256	16.499006	13.058771
S	6.022388	10.996751	13.604365
S	5.924059	0.362436	13.688081
S	13.746077	10.960325	13.763184
S	13.811148	1.202957	13.570889
Au	15.438311	1.249666	5.146704
Au	15.765327	3.891316	5.146704

Au	0.008976	6.601861	5.146704
Au	0.140376	9.310312	5.146704
Au	0.150536	12.063012	5.146704
Au	15.724462	14.846991	5.146704
Au	15.317057	17.668152	5.146704
Au	4.957644	1.052181	5.146704
Au	5.048964	3.794655	5.146704
Au	5.308562	6.59277	5.146704
Au	5.43389	9.412674	5.146704
Au	5.351996	12.255867	5.146704
Au	5.009799	15.054448	5.146704
Au	4.669293	17.886481	5.146704
Au	10.035469	0.949317	5.146704
Au	10.280976	3.656524	5.146704
Au	10.483428	6.433347	5.146704
Au	10.593742	9.239244	5.146704
Au	10.619362	12.069135	5.146704
Au	10.383626	14.894914	5.146704
Au	9.918198	17.685161	5.146704
Au	2.425091	2.426391	5.146704
Au	2.587042	5.242185	5.146704
Au	2.77696	8.031837	5.146704
Au	2.823406	10.814598	5.146704
Au	2.598157	13.588933	5.146704
Au	2.266462	16.348626	5.146704
Au	1.991702	19.058672	5.146704
Au	7.557033	2.273689	5.146704
Au	7.769314	5.088322	5.146704
Au	7.961792	7.939177	5.146704
Au	8.029227	10.781522	5.146704
Au	7.870089	13.590794	5.146704
Au	7.496343	16.312165	5.146704
Au	7.210898	19.036246	5.146704
Au	12.654186	1.926252	5.146704
Au	13.016093	4.710303	5.146704
Au	13.150522	7.53372	5.146704
Au	13.276913	10.315943	5.146704
Au	13.287422	13.075161	5.146704

Au	13.056142	15.805738	5.146704
Au	12.559969	18.464617	5.146704
Au	1.172987	1.941572	7.55819
Au	1.023852	4.904864	7.55819
Au	1.080705	7.786334	7.55819
Au	1.207553	10.639048	7.55819
Au	0.948167	13.577872	7.55819
Au	0.630074	16.614118	7.55819
Au	0.736536	19.768051	7.55819
Au	6.651357	1.061377	7.55819
Au	5.972035	4.80129	7.55819
Au	6.339657	7.893203	7.55819
Au	6.351866	10.95924	7.55819
Au	6.202884	14.045635	7.55819
Au	5.894895	17.025279	7.55819
Au	5.863951	19.938187	7.55819
Au	11.726654	0.58567	7.55819
Au	11.310478	4.892363	7.55819
Au	11.385787	7.831751	7.55819
Au	11.593332	10.732203	7.55819
Au	11.407801	13.647824	7.55819
Au	11.741255	16.615682	7.55819
Au	11.157884	19.363721	7.55819
Au	3.274499	21.221644	7.55819
Au	4.082052	2.373522	7.55819
Au	3.662041	6.458089	7.55819
Au	3.824614	9.409709	7.55819
Au	3.714025	12.433709	7.55819
Au	3.431791	15.395692	7.55819
Au	3.202073	18.346195	7.55819
Au	9.088124	21.252359	7.55819
Au	8.505841	3.383456	7.55819
Au	8.79235	6.340364	7.55819
Au	8.916854	9.393524	7.55819
Au	8.83392	12.399665	7.55819
Au	8.833919	15.40638	7.55819
Au	8.473922	18.329853	7.55819
Au	14.567742	0.160657	7.55819

Au	13.714889	2.972884	7.55819
Au	14.10083	5.870624	7.55819
Au	14.179606	8.725388	7.55819
Au	14.341233	11.582942	7.55819
Au	14.067148	14.550797	7.55819
Au	13.940213	18.834237	7.55819
Au	2.372833	0.483906	10.11201
Au	1.979539	3.411393	9.727763
Au	1.868593	6.230568	9.924643
Au	2.100687	9.128124	9.980411
Au	1.936616	12.250308	9.838682
Au	1.49383	15.047795	9.870514
Au	1.497584	17.800319	9.973551
Au	7.883531	0.335356	10.025673
Au	6.890702	3.117272	9.884418
Au	6.988214	5.998086	9.897441
Au	7.043411	9.743475	9.93577
Au	7.229821	12.591257	9.900022
Au	7.047974	15.461382	9.891394
Au	6.632191	18.286786	9.944391
Au	12.918024	0.678087	10.092824
Au	12.371818	3.437471	9.970257
Au	12.235816	6.256547	9.876885
Au	12.901269	9.843995	9.842987
Au	12.09859	12.639835	10.378515
Au	12.457852	15.311421	9.854102
Au	12.148398	18.134257	9.968219
Au	5.112706	0.319937	9.735035
Au	4.365676	4.872671	9.861293
Au	4.790953	7.845499	9.803525
Au	4.4572	10.965607	9.932623
Au	4.442972	13.889512	10.003462
Au	4.226279	16.693284	9.987373
Au	3.670023	19.5153	10.000284
Au	10.161482	1.813613	9.427344
Au	9.595052	4.635883	9.782714
Au	9.796872	7.803046	9.816821
Au	9.787814	10.942796	9.80666

Au	9.638163	14.032838	9.89269
Au	9.633881	16.798664	9.842106
Au	9.64131	19.640482	9.871472
Au	15.623125	1.422801	9.980057
Au	15.113392	4.245534	9.830125
Au	14.960639	7.15751	9.861894
Au	15.527154	10.654751	9.96542
Au	14.953358	13.493582	9.974907
Au	14.691871	16.956953	9.63246
Au	0.107547	20.224909	10.12068

Comment 9: In the manuscript, the authors state that “This feature further suggested that Cu₇S₄ could be the active site for hydrogen production on Au@Cu₇S₄.”. However, the feasibility of Au@Cu₇S₄ for Hydrogen production does not depend only on the adsorption strength of H atoms, but also on the energy barriers involved in the hydrogen production and release. The evaluation of such barriers relies on the calculation of the corresponding transition state. The authors should mention this aspect.

Response 9: We totally agree with the Reviewer’s perspective that the energy barriers of the transition states involved in the hydrogen production process play decisive role in dictating the overall performance of hydrogen production. We have carried out computations using the nudge elastic band (NEB) approach in order to further resolve these mysteries. However, we are unable to obtain convergence results due to the complex structure of Au@Cu₇S₄. We apologize for this inadequacy.

Comment 10: For the sake of simplicity, I would also suggest to add the ΔG energies to Table 5 of the SI.

Response 10: We have added the ΔG_{H^*} values in the Supplementary Table 5.

Reviewer #2

Comments: This manuscript describes dual-plasmonic Au@Cu₇S₄ yolk@shell nanocrystals for photocatalytic hydrogen production across visible to near infrared spectral region. Authors attribute the facilitated hydrogen production of dual-plasmonic Au@Cu₇S₄ yolk@shell to distinctive absorption in the visible and NIR regions, the advantageous features of yolk@shell nanostructures and the prevalence of long-lived charge separation states prolongs the lifetime of the

delocalized electrons. To justify the charge transfer dynamics, a series of characterizations have been performed. However, several aspects related to the photocatalytic experiments and other linked issues are unclear. I do not recommend the acceptance of this article in Nature communications. In order to justify this decision, I provide to the authors some comments that should be taking into account for its further publication.

Author Response: We thank the Reviewer for critically commenting our work. We highly appreciate the opportunity to further reinforce the manuscript by addressing the questions raised by the Reviewer. We have collected additional data and modified the relevant discussions in the revised manuscript to clarify the Reviewer's concerns. Below are specific responses.

Comment 1: The particle size statistics of all sample void and Au particles should be given to exclude their influence on the performance.

Response 1: We thank the Reviewer for bringing up this important question. The microstructural size of the three Au@Cu₇S₄ and pure Cu₇S₄ had already been specified in the original manuscript. As determined from the TEM observations, the void sizes were 65.7 ± 5.6 nm, 40.0 ± 4.6 nm and 26.5 ± 3.0 nm for 1-Au@Cu₇S₄, 3-Au@Cu₇S₄ and 5-Au@Cu₇S₄, respectively; the void size of pure Cu₇S₄ was 47.1 ± 9.2 nm. The shell thickness of the three Au@Cu₇S₄ and pure Cu₇S₄ was adjusted to a fixed value, which was 11.7 ± 1.5 nm for the three Au@Cu₇S₄ and 11.5 ± 2.3 nm for pure Cu₇S₄. This adjustment allowed the exclusion of the influence of the shell thickness on the photocatalytic properties. On the other hand, we have also examined the particle size distribution of Au for the three Au@Cu₇S₄ and pure Au. The size of the Au yolk of 1-Au@Cu₇S₄, 3-Au@Cu₇S₄ and 5-Au@Cu₇S₄ was 15.3 ± 0.8 nm, 15.2 ± 0.8 nm and 15.2 ± 0.6 nm, respectively; the size of pure Au was 15.2 ± 1.0 nm. The consistency in Au size distribution also excluded its influence on the photocatalytic properties. We have included the above descriptions in the revised manuscript.

Comment 2: Only the post-reaction data of SEM images and XPS spectra cannot prove that the catalyst has considerably high stability, so it is necessary to give the HRTEM, STEM-HAADF, EDS mapping and the particle size statistics of the shell (Cu₇S₄) and the yolk (Au particles).

Response 2: We thank the Reviewer for the attentive examination on this point. In order to confirm the high chemical and structural stability, we have further conducted HRTEM and EDS mapping analysis on 5-Au@Cu₇S₄ after used in hydrogen production for 30 successive hours. As showed in Supplementary Fig. 8, the crystallographic structure and elemental composition of the used 5-Au@Cu₇S₄

remained unchanged. Besides, the void size (27.0 ± 2.6 nm), shell thickness (11.6 ± 0.9 nm) and Au size (15.1 ± 1.0 nm) of the used sample were also examined, showing nearly identical size distribution to that of the as-prepared sample (void size = 26.5 ± 3.0 nm; shell thickness = 11.7 ± 1.5 ; Au size = 15.2 ± 0.6 nm). This outcome corroborated the high chemical and structural stability for Au@Cu₇S₄ toward solar hydrogen production. We have included the above discussions along with Supplementary Fig. 8 in the revised manuscript.

Supplementary Fig. 8 Crystallographic structure and elemental composition of the used 5-Au@Cu₇S₄. **a** HRTEM image, **b** TEM-EDS mapping profiles, **c** TEM image of 5-Au@Cu₇S₄ after used in hydrogen production for 30 successive hours.

Comment 3: There is lack of superiority in the performance of sulphide materials compared to those recent publications, such as Nat Commun 13, 1287 (2022) ; Nat Commun 12, 1343 (2021) ; Nat Commun 13, 4600 (2022) ; Adv. Funct. Mater. 2019, 1901958 ; Adv. Energy Mater. 2016, 1600464 ; Adv. Mater. 2022, 34, 2108475.

Response 3: We thank the Reviewer for bring up this important question. Indeed, there existed many sulfides photocatalysts exhibiting extremely high hydrogen production activities. It should be, however, pointed out that those outstanding sulfides photocatalysts were merely responsive to visible light rather than NIR irradiation. As stated in the original manuscript, there are few choices among the currently available photocatalysts that can respond to NIR irradiation. In comparison with the state-of-the-art NIR-responsive photocatalysts reported so far, the current Au@Cu₇S₄ exhibited an unprecedented, record-breaking quantum yield of 7.3 % at 2200 nm for hydrogen production. On top of that, the hydrogen production activity of Au@Cu₇S₄ toward visible irradiation (AQY = 9.4 % at 500 nm) was also comparable to that of the outstanding sulfides photocatalysts ever reported. Here, we would like to

highlight that the main scope of the current work is to develop a dual plasmonic photocatalyst paradigm that can be responsive to visible and NIR irradiation for realizing wide-spectrum-driven solar hydrogen production. This work is not aimed to create a visible-responsive photocatalyst that can surpass the previously reported sulfides in hydrogen production. The revelation of the remarkable NIR activity of Au@Cu₇S₄ from the current study is exciting and inspiring because it can fill the gap in harvesting the NIR spectrum for the currently available photocatalysts. We have included the above discussions in the revised manuscript. In order to enlighten the readers, we have also introduced the informative references provided by the Reviewer into the performance comparison in Supplementary Table 5.

Comment 4: Is the light intensity of different wavelengths consistent during the in-situ synchrotron radiation test? Besides, the intensity of the corresponding L-edge X-ray absorption spectra of Au and Cu reflects the oxidation state of the corresponding elements. The obtained different intensities of absorption spectra irradiated with different wavelengths of light can only explain the intensity of electron gain and loss of corresponding elements, and it cannot be concluded that "there are three different charge transfer scenarios". Have the authors performed controlled experiments in which the order of different wavelengths of light is varied? The authors mentioned on page 21 that the LSPR wavelength of Au in 5-Au@Cu₇S₄ is between 522.2-755.8 nm, which just corresponds to the visible light region and the infrared region, which means that the light absorption is the strongest in this wavelength range (from Supplementary Fig. 1 can also be concluded that this range has the strongest light absorption), so the synchrotron radiation spectrum shows a large change when irradiated with visible light and infrared light.

Response 4: We thank the Reviewer for raising these critical questions. The proposed three possible charge transfer scenarios were based on the considerations of the band alignment of Au@Cu₇S₄ displayed in Supplementary Fig. 1d. Upon band edge excitation, the upward band bending at the interface facilitated photoexcited hole transfer from Cu₇S₄ to Au, and enabled the photogenerated electrons to be concentrated in Cu₇S₄. Because the photoexcited holes were separated from the photogenerated electrons, radiative electron-hole recombination could be reduced to cause a depressed PL intensity for Au@Cu₇S₄. This contention can be supported by the recorded suppressed PL of Au@Cu₇S₄ shown in Supplementary Fig. 1c. On the other hand, the interfacial upward band bending can also maneuver charge transfer behaviors for hot electrons produced from plasmonic Au and hot holes produced from plasmonic Cu₇S₄. Upon visible irradiation to excite plasmonic Au, the hot electrons can be injected into Cu₇S₄ along the bent conduction band; upon NIR irradiation to

excite plasmonic Cu₇S₄, the hot holes can be passed to Au through the bent valence band. The purpose of conducting in-situ XAS analysis was to further identify these two charge transfer pathways associated with hot electrons and hot holes. The in-situ XAS experiments were performed by firstly applying AM 1.5 G irradiation to allow the three charge transfer pathways to occur. By examining the evolution of UDOS for both Cu₇S₄ and Au components, the accumulation of excited electrons at Cu₇S₄ was identified. As depicted in in Fig. 2c, this outcome can be rationalized by considering the joint operation of the three charge transfer pathways on Au@Cu₇S₄ under AM 1.5 G illumination. In order to decouple the effect of hot electron transfer and derive the influence of hot hole transfer from the joint contribution, the in-situ XAS measurements were further carried out by applying visible irradiation to excite plasmonic Au or by applying NIR irradiation to excite plasmonic Cu₇S₄. Note that visible irradiation was provided by placing a bandpass filter (400-700 nm) over AM 1.5 G irradiation to extract the visible photons from the AM 1.5 G spectrum, giving an irradiation intensity that was 48.2 % of the initial AM 1.5 G irradiation. This would enable a quantitative comparison of UDOS between the results from AM 1.5 G and visible irradiations. The observed greater accumulation of excited electrons at Cu₇S₄ under visible irradiation thus reflected a factual situation caused by the visible photons of the AM 1.5 G spectrum. Since visible irradiation not only excited plasmonic Au, but also caused band edge excitation of Cu₇S₄, two charge transfer events were considered to take place. As depicted in Fig. 2d, the collaborative operation of the two excitations under visible irradiation accounted for the observed greater accumulation of excited electrons at Cu₇S₄. Similarly, NIR irradiation was provided by placing a long-pass filter (> 800 nm) over AM 1.5 G irradiation to extract the NIR photons from the AM 1.5 G spectrum, giving an irradiation intensity that was 54.6 % of the initial AM 1.5 G irradiation. This would again enable a quantitative comparison of UDOS between the results from AM 1.5 G and NIR irradiations. As Fig. 2e shows, NIR irradiation only excited plasmonic Cu₇S₄, and the derived hot hole transfer pathway exclusively accounted for the even larger extent of excited electron accumulation at Cu₇S₄.

Regarding the controlled experiments, we have evaluated the experimental conditions to address this issue. Because in-situ XAS analysis under the three irradiation conditions was performed separately, the order of applying different irradiations did not vary the results. On the other hand, the strong light absorption of Au@Cu₇S₄ at visible and NIR regions did not necessarily give rise to an enlarged extent of UDOS change under visible and NIR irradiations. As mentioned above, the visible and NIR irradiations were provided by placing filters over AM 1.5 G irradiation to separately extract the visible and NIR photons from the AM 1.5 G

spectrum. The intensity of visible and NIR irradiations was therefore substantially lower than that of AM 1.5 G illumination. Despite the much lower irradiation intensity, visible and NIR irradiations still caused a larger extent of UDOS decrease, disclosing the lower efficiency of inducing excited electron accumulation as a result of the interference of the three excitations under AM 1.5 G irradiation. We have included the above discussions in the revised manuscript.

Comment 5: The 5-Au@Cu₇S₄ with the smallest void size has the best performance. If it is further reduced, will the performance be better?

Response 5: We thank the Reviewer for raising this critical question. In fact, it is improbable to obtain Au@Cu₇S₄ with a smaller void size than that of 5-Au@Cu₇S₄ by using the current synthetic method. Note that the void size of Au@Cu₇S₄ was controlled by adjusting the volume of Au colloids employed in the synthesis of the initial Au@Cu₂O template. Addition of more Au colloids gave rise to Au@Cu₂O with a thinner Cu₂O thickness and thereby Au@Cu₇S₄ with a smaller void size. As shown in Supplementary Fig. 3, further reducing the amount of Au colloids to 7.0 mL led to the growth of 7-Au@Cu₇S₄ exhibiting nearly identical structural dimensions to those of 5-Au@Cu₇S₄. If examined closely, the individual 7-Au@Cu₇S₄ contained multiple Au yolks encapsulated in Cu₇S₄ shell, which might be due to the aggregation of Au colloids under the relatively high concentration situation. Since the yolk@shell structural integrity of 7-Au@Cu₇S₄ can no longer be maintained, 7-Au@Cu₇S₄ was not adopted for further performance comparison. Because of this limitation, we are unable to examine if the photocatalytic activity of Au@Cu₇S₄ with an even smaller void size can be better or not. We have included the above discussions along with Supplementary Fig. 3 in the revised manuscript.

Supplementary Fig. 3 Microstructural features of 7-Au@Cu₇S₄. TEM image of 7-Au@Cu₇S₄ prepared by employing 7.0 mL of Au colloids in the synthesis of the initial Au@Cu₂O.

Comment 6: More information on the structure and chemical state of the material should be provided.

Response 6: In the original manuscript, we performed TEM, HRTEM, TEM-EDS, XRD, XPS and XAS analysis to examine the microstructural features and chemical states of the samples. We have also carried out Auger electron spectroscopic analysis on Au@Cu₇S₄ to corroborate the valence state of Cu in the revised manuscript (please refer to Response 7 to Reviewer #2). We believe sufficient information has been acquired from these characterizations.

Comment 7: In Supplementary Fig. 1c on page 21, the author mentioned that XPS proved that the valence state of Cu in the material has always maintained a +2 valence. However, the authors did not give the XPS spectrum of the sample before the reaction. Secondly, the valence state of Cu is difficult to pass through XPS to distinguish +2 and +1 valences, thus it is necessary to provide Auger electron spectra before and after the reaction.

Response 7: We appreciate the Reviewer's concern on this point. The XPS spectra of Cu 2p and S 2p core levels of Au@Cu₇S₄ before photocatalysis had already been provided in the original manuscript. To further corroborate the exclusive existence of Cu¹⁺, we have carried out Auger electron spectroscopic analysis on Au@Cu₇S₄ before and after photocatalysis. As shown in Supplementary Fig. 7e, the Cu LMM spectrum

for Au@Cu₇S₄ before and after photocatalysis was nearly identical, exhibiting a peak centered around 916.4 to 916.6 eV. This kinetic energy can be convincingly assigned to Cu¹⁺.¹¹ We have included the above discussions and Supplementary Fig. 7e in the revised manuscript.

Supplementary Fig. 7 Stability tests on 5-Au@Cu₇S₄ for solar hydrogen production. SEM image of 5-Au@Cu₇S₄ **a** before and **b** after used in hydrogen production for 30 successive hours. Corresponding XPS spectra of **c** Cu 2p and **d** S 2p core levels, and **e** Auger spectra of Cu LMM.

Comment 8: The determination of the active site is only through theoretical calculations. Is there any corresponding experimental evidence? Besides, the established theoretical model does not correspond to the experiment.

Response 8: From the results of DFT calculations, we proposed that Cu₇S₄ could be the active site for hydrogen production on Au@Cu₇S₄. The experimental observations on the charge transfer scenarios can provide complementary support to this proposition. As illustrated in Fig. 5, under visible irradiation, the photoexcited holes of Cu₇S₄ would be favorably transported to Au, leaving photogenerated electrons at Cu₇S₄ to reduce protons for hydrogen evolution. The hot electrons of plasmonic Au can also be injected into Cu₇S₄ to contribute to hydrogen production. Under NIR irradiation, the hot holes of plasmonic Cu₇S₄ can preferentially transfer to Au. The delocalized hot electrons at Cu₇S₄ can then reduce protons to evolve hydrogen. These observations all pointed out the fact that Cu₇S₄ could function as a favourable site for conducting hydrogen production on Au@Cu₇S₄, consistent with the results of DFT calculations.

Indeed, the structural models used for DFT calculations did not completely equate to the microstructural features of the samples in terms of the structural dimensions. For DFT calculations, only a few atomic layers can be considered for modelling in order to receive convergence results. Nevertheless, the computed data can deliver atomic-scale insights into the thermodynamics and kinetics properties at the surficial region. This information is particularly relevant to the fundamental understanding of catalytic mechanism since catalytic reactions mostly occur at the localized surface of catalysts.^{3, 4, 5} We have included the above discussions in the revised manuscript.

Comment 9: Through a series of experiments, the authors prove that the material has excellent photogenerated carrier separation and inhibited recombination capabilities in the infrared and visible light regions. It is not optimal under AM1.5, why is the performance test under AM1.5? Secondly, the authors did not rule out the thermal effect during the test.

Response 9: AM 1.5 G illumination represents a specification commonly used in photocatalysis community to evaluate the activity of hydrogen production, enabling global activity comparison with other benchmark photocatalysts ever reported. To highlight the remarkable NIR activity, we also examined the AQYs of hydrogen production across the NIR region. As summarized in Supplementary Table 3, compared with the state-of-the-art NIR-responsive photocatalysts reported so far, the

current Au@Cu₇S₄ exhibited an unprecedented, record-breaking quantum yield of 7.3 % at 2200 nm for hydrogen production.

On the other hand, the thermal effect induced by light irradiation has also been investigated by recording the temperature profiles of the electrolyte containing 5-Au@Cu₇S₄ before and after for 30 successive hours of AM 1.5 G illumination. As Supplementary Fig. 9 reveals, upon 30 successive hours of light irradiation, the temperature of the Au@Cu₇S₄-contained electrolyte increased from 27.1 to 28.4 °C, whereas the temperature of pure electrolyte increased from 27.2 to 27.6 °C during the same irradiation period. Such a minimal temperature rise was far less than the temperature required for proceeding with photochemical hydrogen production, even when a sophisticated catalyst is present (higher than 100 °C).^{12,13} With this observation, the thermal effect on hydrogen production can be neglected in the current system. We have included the above discussions and Supplementary Fig. 9 in the revised manuscript.

Supplementary Fig. 9 Thermal effect induced by light irradiation. Temperature profiles of the electrolyte containing 5-Au@Cu₇S₄ **a** before and **b** after for 30 successive hours of AM 1.5 G illumination. **c** and **d** show the corresponding temperature profiles of pure electrolyte. The temperature was recorded at five positions along the vertical direction of the vessel. An averaged value was then present.

Comment 10: Please supplement electrochemical impedance spectroscopy, photocurrent spectroscopy, and steady-state fluorescence spectroscopy to illustrate the optoelectronic properties of materials.

Response 10: We agree with the Reviewer that electrochemical impedance spectroscopy (EIS) and photocurrent spectroscopy may provide further informative data to investigate the optoelectronic properties of the samples. These data are particularly relevant to the interpretation of the (photo)electrochemical behaviors when the samples are employed as (photo)electrodes. Since the current Au@Cu₇S₄ was employed as photocatalyst powers rather than (photo)electrodes, limited information relevant to the photocatalytic properties can be learned from EIS and photocurrent data. We therefore did not perform these measurements. On the other hand, the steady-state fluorescence spectroscopy is indeed important to photocatalysts because it can be used to study the charge transfer behaviors. These data were already provided in Supplementary Fig. 1 in the original manuscript.

Reviewer #3

Comments: The authors reported that the Au@Cu₇S₄ yolk@shell nanostructure as a dual-plasmonic photocatalyst to achieve remarkable hydrogen production under visible and NIR illumination, including the synthesis, microstructural, LSPR property and charge transfer dynamics of the nanostructures. The results indicated that the Au@Cu₇S₄ exhibits superior hydrogen production. I think that the results of this manuscript should be interesting to the readership of Nature Communications. However, the following questions should be properly addressed by the authors before this manuscript can be accepted for publication:

Author Response: We thank the Reviewer for the positive comments. We have collected additional data and modified the relevant discussions in the revised manuscript to address the Reviewer's questions. Below are specific responses.

Comment 1: The author claims that the Au@Cu₇S₄ nanostructures comprised a movable Au particle surrounded by a hollow Cu₇S₄. As Cu₇S₄ and Au are brought in contact, whether the band bending is related to the chemical bonding behavior between Au particles and Cu₇S₄, and is interrelated Au particles are in inside or outside of Cu₇S₄ shell.

Response 1: We thank the Reviewer for bringing up this important question. As witnessed by real-time TEM observations,^{14,15} the Au yolk inside the Cu₇S₄ shell can freely move, especially when the individual yolk@shell nanocrystals were suspended in the solution. This feature suggested that Au yolk was not chemically bound to

Cu₇S₄ shell. Even though Au and Cu₇S₄ were not chemically bonded, the frequent collision of Au with Cu₇S₄ in the suspension state still induced electronic interactions to enable effective interfacial charge transfer as demonstrated from TAS measurements. On the other hand, as noticed from the TEM images in Fig 1(a-c), the as-prepared Au@Cu₇S₄ exhibited high structural integrity of yolk@shell architecture without the existence of un-coated Au outside the Cu₇S₄ shell. The interfacial charge transfer was thus considered to occur at the interior of Au@Cu₇S₄.

Comment 2: Hot electron-hole pairs generated by LSPR have an important impact for photocatalytic hydrogen production, please briefly describe the effect of Au/Cu₇S₄ interface and the possible mechanism.

Response 2: We thank the Reviewer for the attentive examination on this point. As displayed in Fig. 5, under visible irradiation, both band edge excitation of Cu₇S₄ and plasmonic excitation of Au occurred, producing photoexcited charge carriers and hot electrons, respectively. Because of the upward band bending at the Au/Cu₇S₄ interface, the photoexcited holes of Cu₇S₄ would be favorably transported to Au through the bent valence band. The photogenerated electrons would then be concentrated at Cu₇S₄ to reduce protons for hydrogen evolution. On the other hand, the hot electrons of plasmonic Au were highly energetic, which can overcome the barrier of the bent conduction band of Cu₇S₄ to be injected into Cu₇S₄.¹⁶ These hot electrons contributed to hydrogen production as well. Under NIR irradiation, the hot holes of plasmonic Cu₇S₄ can preferentially transfer to Au through the bent valence band. The delocalized hot electrons at Cu₇S₄ can then reduce protons to evolve hydrogen. We have included the above discussions in the revised manuscript.

Comment 3: Au colloids and pure Cu₇S₄ suspension was simply mixed and denoted as Au+Cu₇S₄, as a comparison, Au+Cu₇S₄ and Au@Cu₇S₄ yolk@shell nanocrystals are incomparable since their density is different.

Response 3: We appreciate the Reviewer's concern on this point. As stated in the Supplementary methods section, Au+Cu₇S₄ was obtained by simply mixing Au colloids and pure Cu₇S₄ suspension with their concentrations deliberately adjusted to equate those of 5-Au@Cu₇S₄. In other words, the amount of Au and Cu₇S₄ in Au+Cu₇S₄ was respectively equal to the amount of Au and Cu₇S₄ in 5-Au@Cu₇S₄. With this deliberate control can the activity performance comparison of Au+Cu₇S₄ with Au@Cu₇S₄ give a reliable conclusion that yolk@shell nanostructures played a critical role in achieving superior hydrogen production. We have added the above discussions in the revised manuscript.

References

1. Nørskov J.K., *et al.* Trends in the exchange current for hydrogen evolution. *J. Electrochem. Soc.* **152**, J23 (2005).
2. Feng, J.-X., Wu, J.-Q., Tong, Y.-X., Li, G.-R. Efficient hydrogen evolution on Cu nanodots-decorated Ni₃S₂ nanotubes by optimizing atomic hydrogen adsorption and desorption. *J. Am. Chem. Soc.* **140**, 610-617 (2018).
3. Gao, D., Xu, J., Wang, L., Zhu, B., Yu, H., Yu, J. Optimizing atomic hydrogen desorption of sulfur-rich NiS_{1+x} cocatalyst for boosting photocatalytic H₂ evolution. *Adv. Mater.* **34**, 2108475 (2022).
4. Zhong, W., Gao, D., Yu, H., Fan, J., Yu, J. Novel amorphous NiCuS_x H₂-evolution cocatalyst: Optimizing surface hydrogen desorption for efficient photocatalytic activity. *Chem. Eng. J.* **419**, 129652 (2021).
5. Huang, S., *et al.* Amorphous NiWO₄ nanoparticles boosting the alkaline hydrogen evolution performance of Ni₃S₂ electrocatalysts. *Appl. Catal. B* **274**, 119120 (2020).
6. Shimizu, F., Ogata, S., Li, J. Theory of shear banding in metallic glasses and molecular dynamics calculations. *Mater. Trans.* **48**, 2923-2927 (2007).
7. Patra, A.S., *et al.* Photocatalytic activity enhancement of Cu₂O cubes functionalized with 2-ethynyl-6-methoxynaphthalene through band structure modulation. *J. Mater. Chem. C* **10**, 3980-3989 (2022).
8. Kao, J.-C., *et al.* Electron injection via interfacial atomic Au clusters substantially enhance the visible-light-driven photocatalytic H₂ production of the PF3T enclosed TiO₂ nanocomposite. *Small* **n/a**, 2303391 (2023).
9. Zhou, C., Gao, J., Deng, Y., Wang, M., Li, D., Xia, C. Electric double layer-mediated polarization field for optimizing photogenerated carrier dynamics and thermodynamics. *Nat. Commun.* **14**, 3592 (2023).
10. Martínez-Alonso, C., Guevara-Vela, J.M., Llorca, J. Understanding the effect of mechanical strains on the catalytic activity of transition metals. *Phys. Chem.*

Chem. Phys. **24**, 4832-4842 (2022).

11. Shaaban, E., Li, G. Probing active sites for carbon oxides hydrogenation on Cu/TiO₂ using infrared spectroscopy. *Commun. Chem.* **5**, 32 (2022).
12. Docao, S., Koirala, A.R., Kim, M.G., Hwang, I.C., Song, M.K., Yoon, K.B. Solar photochemical–thermal water splitting at 140 °C with Cu-loaded TiO₂. *Energy Environ. Sci.* **10**, 628-640 (2017).
13. Rao, C.N.R., Dey, S. Solar thermochemical splitting of water to generate hydrogen. *PNAS* **114**, 13385-13393 (2017).
14. Wu, J.-Y., *et al.* Electronic interactions and charge-transfer dynamics for a series of yolk–shell nanocrystals: Implications for photocatalysis. *ACS Appl. Nano Mater.* **5**, 8404-8416 (2022).
15. NOD Lab at NCTUMSE. Movement of the Au yolk inside the Cu₇S₄ shell for Au@Cu₇S₄ yolk-shell nanocrystals. YouTube https://www.youtube.com/watch?v=_cgUeqesgDs (2022).
16. Won, R. Ultrafast atomic probe. *Nat. Photonics* **7**, 85-85 (2013).

REVIEWER COMMENTS

Reviewer #1 (Remarks to the Author):

In their "response to referee" file, the authors write that "the main scope of the current work is experimentally developing a dual plasmonic photocatalyst paradigm that can be responsive to visible and NIR irradiation for realizing solar hydrogen production. This work is not aimed to create a theoretical approach to the computational study of the photocatalytic mechanism."

However, I would like to point out that whatever is the goal of any study, details describing how experiments and calculations are performed should always be added. On the other hand, in this revised version the authors have sufficiently addressed all my comments and covered the severe missing information of the previous version. Based on that, I believe the current version of the manuscript is now suitable for publication in Nature Communication

Reviewer #2 (Remarks to the Author):

The authors have answered my questions well, and I suggest to accept this paper in Nature Communication

Reviewer #3 (Remarks to the Author):

The author has made a careful correction for this manuscript according to our suggestions. These responses to our questions are reasonable and satisfactory. The overall level of this article has been greatly improved. So I recommend the version to publish in the journal of Nature Communications.

Reviewer #4 (Remarks to the Author):

In this manuscript, Au@Cu₇S₄ yolk@shell nanocrystals as dual-plasmonic photocatalysts were synthesized and used for photocatalytic hydrogen release from aqueous electrolyte containing 5.0 vol.% methanol and 15.0 wt.% glucose. Although the authors have symmetrically characterized the materials and analyzed the photocatalytic reaction mechanism, I recommend rejecting the manuscript because the key data lacks credibility and novelty.

Au@Cu_{2-x}S dual-plasmonic photocatalysts have been extensively investigated. This manuscript did not present any new results except using In-situ XAS to investigate charge transfer dynamics. More importantly, the work mistakenly ignores the contribution of photothermal effect. The authors did not use cooling water to control the temperature of the reaction solution in the photocatalysis process, and hence the temperature of the reaction solution should increase quickly because of the LSPR effect and also the heating effect of the NIR light. In this work, the authors stated that the temperature of the 5-Au@Cu₇S₄-contained electrolyte increased from 27.1 to 28.4 °C and hence the thermal effect on hydrogen production has been neglected. Actually, the photothermal effect of Au@Cu_{2-x}S have been widely studied and recognized. Here, I only give some examples.

In the minireview (Chem. Eur. J. 2021, 27, 11030-11040), titled "Dual Plasmonic Au-Cu_{2-x}S Nanocomposites: Design Strategies and Photothermal Properties", the tunable plasmon-induced absorption features and photothermal properties were symmetrically introduced. Core-shell Au@CuS show higher temperature rise in comparison with single Au and CuS. Because of the outstanding photothermal effect of core-shell Au@CuS, the temperature of the aqueous solution can be quickly increased for example to 65 °C under Xe lamp (J. Mater. Chem. A, 2019, 7, 3408-3414,). The Au@CuS core-shell NPs exhibit photothermal property under ultraviolet, visible and NIR light irradiation, and hence increase the temperature of the aqueous solution substantially (ACS Appl. Mater. Interfaces 2020, 12, 46146-46161, ChemPhysChem 2018, 19, 1852 - 1858).

In summary, a large number of literatures prove that Au@Cu_{2-x}S has a strong photothermal effect,

which can significantly improve the photocatalytic activity. The work in this manuscript did not provide substantive new results for the study of photocatalysis of the Au@Cu₂-xS system. More seriously, the work mistakenly ignored the contribution of photothermal effect for the photocatalytic performance. All in all, this paper did not meet the requirement of the prestigious Nature Communications.

Summary of changes and replies to the Reviewers' comments

Reviewer #1

Comments: In their "response to referee" file, the authors write that "the main scope of the current work is experimentally developing a dual plasmonic photocatalyst paradigm that can be responsive to visible and NIR irradiation for realizing solar hydrogen production. This work is not aimed to create a theoretical approach to the computational study of the photocatalytic mechanism." However, I would like to point out that whatever is the goal of any study, details describing how experiments and calculations are performed should always be added. On the other hand, in this revised version the authors have sufficiently addressed all my comments and covered the severe missing information of the previous version. Based on that, I believe the current version of the manuscript is now suitable for publication in Nature Communication.

Author Response: We thank the Reviewer for giving the positive, righteous comments on the revised manuscript. We also thank the Reviewer for the appreciation of our work.

Reviewer #2

Comments: The authors have answered my questions well, and I suggest to accept this paper in Nature Communication.

Author Response: We thank the Reviewer for the appreciation of our work.

Reviewer #3

Comments: The author has made a careful correction for this manuscript according to our suggestions. These responses to our questions are reasonable and satisfactory. The overall level of this article has been greatly improved. So I recommend the version to publish in the journal of Nature Communications.

Author Response: We thank the Reviewer for the appreciation of our work.

Reviewer #4

Comments: In this manuscript, Au@Cu₇S₄ yolk@shell nanocrystals as dual-plasmonic photocatalysts were synthesized and used for photocatalytic hydrogen release from aqueous electrolyte containing 5.0 vol.% methanol and 15.0 wt.% glucose. Although the authors have symmetrically characterized the materials and analyzed the photocatalytic reaction mechanism, I recommend rejecting the manuscript because the key data lacks credibility and novelty. Au@Cu_{2-x}S dual-plasmonic photocatalysts have been extensively investigated. This manuscript

did not present any new results except using In-situ XAS to investigate charge transfer dynamics. More importantly, the work mistakenly ignores the contribution of photothermal effect. The authors did not use cooling water to control the temperature of the reaction solution in the photocatalysis process, and hence the temperature of the reaction solution should increase quickly because of the LPSR effect and also the heating effect of the NIR light. In this work, the authors stated that the temperature of the 5-Au@Cu₇S₄-contained electrolyte increased from 27.1 to 28.4 °C and hence the thermal effect on hydrogen production has been neglected. Actually, the photothermal effect of Au@Cu_{2-x}S have been widely studied and recognized. Here, I only give some examples. In the minireview (Chem. Eur. J. 2021, 27, 11030-11040), titled “Dual Plasmonic Au–Cu_{2-x}S Nanocomposites: Design Strategies and Photothermal Properties”, the tunable plasmon-induced absorption features and photothermal properties were symmetrically introduced. Core–shell Au@CuS show higher temperature rise in comparison with single Au and CuS. Because of the outstanding photothermal effect of core–shell Au@CuS, the temperature of the aqueous solution can be quickly increased for example to 65 °C under Xe lamp (J. Mater. Chem. A, 2019, 7, 3408–3414,). The Au@CuS core–shell NPs exhibit photothermal property under ultraviolet, visible and NIR light irradiation, and hence increase the temperature of the aqueous solution substantially (ACS Appl. Mater. Interfaces 2020, 12, 46146–46161, ChemPhysChem 2018, 19, 1852 – 1858). In summary, a large number of literatures prove that Au@Cu_{2-x}S has a strong photothermal effect, which can significantly improve the photocatalytic activity. The work in this manuscript did not provide substantive new results for the study of photocatalysis of the Au@Cu_{2-x}S system. More seriously, the work mistakenly ignored the contribution of photothermal effect for the photocatalytic performance. All in all, this paper did not meet the requirement of the prestigious Nature Communications.

Author Response: We thank the Reviewer for the favorable review of our manuscript and for raising insightful questions. First of all, we would like to emphasize that we did not exclude the existence of photothermal effect at the current Au@Cu₇S₄. In the original manuscript, we suggested that the thermal effect on hydrogen production can be neglected because the temperature rise induced by light irradiation was insufficient (around 1.5 °C) for facilitating the kinetics of hydrogen production reaction. Previous study also suggested that the thermal energy induced by light irradiation could barely promote photocatalytic reactions in a liquid-solid heterogeneous system due to the rapid heat dissipation to the surrounding liquid.¹ To validate this contention, we have performed additional photocatalytic experiments on 5-Au@Cu₇S₄ by controlling the electrolyte temperature. As displayed in Supplementary Fig. 10, by introducing a cold plate, the electrolyte temperature can maintain a nearly constant value throughout the

photocatalytic reaction process. The resultant hydrogen production performance was compared with that obtained without temperature control. The difference in hydrogen production rate between the two conditions (under temperature control and without temperature control) was 3.6 % based on the results of four duplicate sets of experiments. This outcome validated that the thermal effect on hydrogen production can be considered rather minor in the current photocatalytic system. This suggestion, however, did not imply the neglect of photothermal effect itself. In fact, the current Au@Cu₇S₄ is anticipated to exhibit photothermal effect since both Au and Cu₇S₄ are capable of depositing thermal energy to lattice vibrations upon LSPR excitation. Nevertheless, the observed temperature rise for 5-Au@Cu₇S₄-contained electrolyte was fairly limited. The cause was believed to be associated with the considerably large volume of the electrolyte (40.0 mL) and relatively low power of light irradiation (100 mW cm⁻²) employed in photocatalytic reactions. Note that photothermal effect in terms of a substantial rise in solution temperature can only be experimentally probed when the sample is dispersed in a solvent of small volume (from hundreds of μL to a few mL) and irradiated with a laser of high power (from a few W cm⁻² to tens of W cm⁻²).²⁻⁷ In a previous study, an noticeable temperature increase to around 40 °C was observed for a 2.0 mL aqueous solution containing Au@Cu_{1.5}S core@shell nanoparticles under laser irradiation (λ = 980 nm, power = 6.25 W cm⁻²).³ In the other study, a pronounced temperature elevation to 52.2 °C was recorded on a 1.0 mL aqueous dispersion of Au nanorod@Cu₇S₄ nanooctahedron yolk@shell particles upon laser irradiation (λ = 915 nm, power = 2.12 W cm⁻²).² In a different work, a substantial temperature rise to 72.9 °C was achieved by irradiating Au@Cu_{2-x}S core@shell nanorod suspension with light covering the whole infrared spectrum (λ = 700-2000 nm, power = 1 W cm⁻²).⁴ These demonstrations showed that a small volume of sample solution accompanied with an intense laser irradiation is required to achieving noticeable temperature increase caused by photothermal heating. This requirement is particularly indispensable to plasmonic materials because plasmonic heating is localized at the sample surface and is only effective in the vicinity of sample surface. If a plasmonic material is dispersed in a large medium and irradiated by a weak light, the generated thermal energy is limited and cannot effectively heat up the whole surrounding medium. This explained why a minimal temperature rise was observed for 5-Au@Cu₇S₄-contained electrolyte in the current photocatalytic system, in which a large electrolyte volume and a low irradiation power were set.

Supplementary Fig. 10 Thermal effect with and without temperature control.

Temperature profiles of the electrolyte containing 5-Au@Cu₇S₄ under temperature control **a** before, **b** after for 1 hour and **c** after 6 hours of AM 1.5 G illumination. **d**, **e** and **f** show the corresponding temperature profiles of the same electrolyte without temperature control. The temperature was recorded at five positions along the vertical direction of the vessel. An averaged value was then present. The higher background temperature of **d**, **e**, **f** than that of **a**, **b**, **c** resulted from the heat generated by the cold plate under operation, which can be confirmed in **Supplementary Fig.11**. Nevertheless, the electrolyte temperature can be well-controlled by the cold plate.

Supplementary Fig. 11 Background temperature disturbance by cold plate.

Temperature profiles of pure electrolyte in dark **a** as the vessel was placed on a plain plate without temperature control, and **b** as the vessel was placed on a cold plate under temperature control. As the cold plate operated, it generated heat to cause a slight increase in background temperature.

Secondly, to examine the data credibility of thermal effect, we have recorded the temperature profiles of the electrolytes containing pure Cu₇S₄ or pure Au under AM 1.5 G illumination without temperature control. Supplementary Figs. 12-15 summarize the temperature profiles for all of the electrolytes taken at a given time interval of irradiation. The electrolytes containing pure Au or pure Cu₇S₄ also

exhibited a fairly limited temperature increase (less than 1.5 °C) upon 30 successive hours of light irradiation. This outcome was expectable because of the employment of a large volume of electrolyte and a low power of irradiation. The consistency of the recorded temperature profiles for all of the electrolytes further ensured the reproducibility of the experiments. In order to highlight the relevance of irradiation power, we have also recorded the temperature profiles of 5-Au@Cu₇S₄-contained electrolyte under two-sun (200 mW cm⁻²) and three-sun (300 mW cm⁻²) irradiation conditions. As Supplementary Fig. 16 compares, the temperature rise of 5-Au@Cu₇S₄-contained electrolyte upon 6 hours of one-sun, two-sun and three-sun irradiation respectively reached 1.5, 2.8 and 4.5 °C. The accordingly increased extent of temperature rise with increasing irradiation power revealed the importance of irradiation power in heating up the whole electrolyte surrounding Au@Cu₇S₄. This finding also stood up for the argument that a small volume of sample solution accompanied with an intense laser irradiation is required to achieving noticeable temperature increase caused by photothermal heating.

Supplementary Fig. 12 Thermal effect induced by light irradiation on Au@Cu₇S₄.

Temperature profiles of the electrolyte containing 5-Au@Cu₇S₄ taken at a given time interval of AM 1.5 G illumination. The temperature was recorded at five positions along the vertical direction of the vessel. An averaged value was then present.

Supplementary Fig. 13 Thermal effect induced by light irradiation on pure Au. Temperature profiles of the electrolyte containing pure Au taken at a given time interval of AM 1.5 G illumination. The temperature was recorded at five positions along the vertical direction of the vessel. An averaged value was then present.

Supplementary Fig. 14 Thermal effect induced by light irradiation on pure Cu₇S₄. Temperature profiles of the electrolyte containing pure Cu₇S₄ taken at a given time interval of AM 1.5 G illumination. The temperature was recorded at five positions along the vertical direction of the vessel. An averaged value was then present.

Supplementary Fig. 15 Thermal effect induced by light irradiation on pure electrolyte. Temperature profiles of pure electrolyte taken at a given time interval of AM 1.5 illumination. The temperature was recorded at five positions along the vertical direction of the vessel. An averaged value was then present.

Supplementary Fig. 16 Effect of irradiation power on temperature rise for Au@Cu₇S₄. Temperature profiles of the electrolyte containing 5-Au@Cu₇S₄ taken at a given time interval under different irradiation conditions: a, b, c one-sun irradiation (100 mW cm⁻²); d, e, f two-sun irradiation (200 mW cm⁻²); g, h, i three-sun irradiation (300 mW cm⁻²). The temperature was recorded at five positions along the vertical direction of the vessel. An averaged value was then

present.

Thirdly, for the current Au@Cu₇S₄, the temperature rise of the electrolyte induced by photothermal heating can reach a level approximating to those of Au and Cu_{2-x}S-based heterostructures reported in the literature (from 40 to 72.9 °C),²⁻⁴ provided that the required experimental conditions, i.e. electrolyte volume and irradiation power, are applied. To examine the intrinsic features of photothermal heating, we have conducted new photothermal experiments by irradiating 0.2 mL of 5-Au@Cu₇S₄-contained electrolyte with high-power lasers under various excitation wavelengths from visible to NIR region. Supplementary Fig. 17 displays the resultant temperature evolutions under various irradiation conditions. Both visible ($\lambda = 532$ nm and 650 nm) and NIR excitations ($\lambda = 785$ nm, 808 nm, and 1064 nm) caused a perceivable, gradual temperature increase with irradiation time. The temperature rising processes can be visualized from the corresponding thermograph images showed in Supplementary Fig. 18. Notably, under 808 nm laser excitation (power = 2.0 W cm⁻²) for 20 min, the electrolyte temperature of 5-Au@Cu₇S₄ can exceed 57 °C as a result of the prevalence of pronounced photothermal effect. This observation further corroborated the argument that a small volume of sample solution accompanied with an intense laser irradiation is required to achieving noticeable temperature increase caused by photothermal heating. It is important to note that the achievable temperature rise for photothermal materials is highly sensitive to the experimental conditions,^{3,5,6,8} such as the concentration of the dispersed photothermal materials, the wavelength and power of the irradiation, as well as the volume of the solution. As an illustration, the photothermal performance of 5-Au@Cu₇S₄-contained electrolyte (0.2 mL) has also been examined under 808 nm irradiation with a power set to be equal to the irradiance of the standard AM 1.5 G spectra (0.11 mW cm⁻² at 808 nm as determined from ASTM G-173093 data set). As showed in Supplementary Fig. 19, the electrolyte temperature nearly unchanged despite the use of a small electrolyte volume. Additional comparative experiment has further been performed by irradiating 5-Au@Cu₇S₄-contained electrolyte (0.2 mL) at an irradiation wavelength where standard AM 1.5 G spectra show peak irradiances (around 0.15 mW cm⁻² at 440 nm). The electrolyte temperature did not vary as well. These illustrations reflected that the visible and NIR photons of AM 1.5 G illumination were essentially ineffective for causing a noticeable temperature rise for 5-Au@Cu₇S₄ induced by photothermal heating even though a small volume of electrolyte was employed. This observation also supported our explanations on the cause for the minimal temperature rise observed for 5-Au@Cu₇S₄-contained electrolyte in the current photocatalytic system, in which a large electrolyte volume and a low irradiation power were set. The

dependence of photothermal performance on experimental conditions has made the direct comparison of the temperature rise of the current Au@Cu₇S₄ with those reported in other Cu_{2-x}S-based photothermal systems improbable. We believe there is definitely room for the further improvement of photothermal performance for the current Au@Cu₇S₄ by optimizing the experimental factors. These tasks are important, but fall behind the scope of the present work.

Supplementary Fig. 17 Photothermal heating upon Au@Cu₇S₄. Temperature evolution of 5-Au@Cu₇S₄-contained electrolyte (0.2 mL) under laser excitation at **a** 532 nm (2.0 W cm⁻²), **b** 650 nm (1.0 W cm⁻²), **c** 785 nm (4.0 W cm⁻²), **d** 808 nm (2.0 W cm⁻²), and **e** 1064 nm (0.33 W cm⁻²). The power of irradiation was adjusted to the maximal capacity of the laser in order to highlight the photothermal effect. The rise of electrolyte temperature of 5-Au@Cu₇S₄ under 1064 nm irradiation was less pronounced than that recorded under 808 nm irradiation because of the lower laser power employed. The results of pure electrolyte were also included.

Supplementary Fig. 18 Photothermal heating upon Au@Cu₇S₄. Thermograph images of 5-Au@Cu₇S₄-contained electrolyte (volume = 0.2 mL) in a vial (capacity = 0.4 mL) taken at a given time interval under laser excitation at **a** 532 nm (2.0 W cm⁻²), **b** 650 nm (1.0 W cm⁻²), **c** 785 nm (4.0 W cm⁻²), **d** 808 nm (2.0 W cm⁻²), and **e** 1064 nm (0.33 W cm⁻²). The power of irradiation was adjusted to the maximal capacity of the laser in order to highlight the photothermal effect.

Supplementary Fig. 19 Effectiveness of AM 1.5 G illumination in photothermal heating upon Au@Cu₇S₄. Temperature evolution of 5-Au@Cu₇S₄-contained electrolyte (0.2 mL) under laser excitation at 470 nm (0.15 mW cm⁻²) and 808 nm (0.11 mW cm⁻²) with the power set to be equal to the irradiance of the standard AM 1.5 G spectra (ASTM G-173093 data set).

In recent years, dual-plasmonic heterostructures comprising plasmonic metals and plasmonic semiconductors have been widely investigated due to the intriguing optical properties resulting from the synergy of the two LSPR features.^{9, 10} Previous studies have demonstrated the extensive use of dual-plasmonic heterostructures in photothermal and biomedical applications. For photocatalytic applications, using dual-plasmonic heterostructures as photocatalysts is still in its infancy. Supplementary Table 7 summarizes the recent development of dual-plasmonic photocatalysts and the scenarios of their photocatalytic applications. Most of the reaction scenarios ever demonstrated on dual-plasmonic photocatalysts were the degradation of organic dyes. For NIR-driven hydrogen production, only one practice has been made on Au/CuSe tangential hybrids, showing 0.34 % of AQY of hydrogen production at 940 nm irradiation in the presence of Pt co-catalyst.¹¹ As a comparison, the current Au@Cu₇S₄ has achieved an advanced AQY of 2.7 % at 900 nm and a record-breaking AQY of 7.3 % at 2200 nm in the absence of additional co-catalysts. This achievement has never been realized by previously reported dual-plasmonic photocatalysts and has surpassed the performance of other state-of-the-art NIR-responsive photocatalysts. As stated in the Conclusion section of the original manuscript, the revelation of the remarkable NIR activity of Au@Cu₇S₄ is exciting and inspiring because it can fill the gap in harvesting the NIR spectrum for the currently available photocatalysts. We believe these findings are sufficiently novel and substantially impactful to meet the high criterion of being considered by *Nature Communications*. In a short summary, we have included all of the above discussions and new experimental data along with supporting references in the revised manuscript. With these revisions and justifications, we sincerely hope the Reviewer can re-consider our work for

publication.

Supplementary Table 7. Comparison of photocatalytic performance and reaction scenarios with dual-plasmonic metal-semiconductor photocatalysts reported in the literature.

Photocatalyst Composition [Heterostructure Type] [Co-catalyst]	Reaction Scenario	Active Region	Activity	Ref.
Au@Cu ₇ S ₄ [yolk@shell] [none]	photocatalytic hydrogen production	AM 1.5 G	yield = 211 $\mu\text{mol h}^{-1} \text{g}^{-1}$ AQY = 9.4 % at 500 nm, 2.7 % at 900 nm, 7.3 % at 2200 nm	This work
Au/CuSe [tangential] [Pt]	photocatalytic hydrogen production	$\lambda > 420 \text{ nm}$	yield = 4180 $\mu\text{mol h}^{-1} \text{g}^{-1}$ AQY = 0.30 % at 500nm, 0.34 % at 940 nm	11
Au-Cu _{2-x} Te [disk-on-dot] [none]	photoelectrochemical hydrogen production	white light	photocurrent = 2.37 mA cm^{-2} at -0.4 V _{RHE} AQY = not available	12

Au@Cu _{2-x} Se	photocatalytic		rate constant = 0.23 min ⁻¹ at λ > 420 nm, 0.13 min ⁻¹ at λ > 760 nm	13
[eccentric]	degradation of	λ > 420 nm		
[none]	rhodamine B		AQY = not available	
Au@CuS	photocatalytic		rate constant = 0.012 min ⁻¹ at 445 nm, 0.009 min ⁻¹ at 638 nm, 0.008 min ⁻¹ at 980 nm	3
[core@shell]	degradation of	visible to NIR		
[none]	rhodamine B		AQY = not available	
Au@WO _{3-x}	photocatalytic		^a yield = 283 μmol h ⁻¹ g ⁻¹	14
[core@shell]	hydrolysis of	λ > 420 nm		
[none]	ammonia borane		AQY = not available	
Au/CdS-Cu _{2-x} S	photocatalytic		rate constant = not available	7
[core/shell]	degradation of	λ > 420 nm	^b AQY = 1.68 × 10 ⁻¹² mol mW ⁻¹ h ⁻¹	
[none]	rhodamine B			
Au/Cu _{2-x} S	photocatalytic	λ > 420 nm	rate constant = 0.072 min ⁻¹	15

[half-shell] [none]	degradation of methylene blue		AQY = not available	
CuS/Au [nanoplate/nanoparticle] [none]	photocatalytic degradation of methylene blue	300-1400 nm	rate constant = 0.37 min ⁻¹ AQY = not available	16
W ₁₈ O ₄₉ -Au [nanobundle-nanoparticle] [none]	photocatalytic nitrophenol hydrogenation	white light	rate constant = not available AQY = not available	17

^ayield was calculated by the authors by using relevant data provided in the paper.

^bAQY of rhodamine B degradation was defined as the percentage of the rate of concentration change of rhodamine B to the power of irradiation.

References

1. Huang, H., Shi, R., Zhang, X., Zhao, J. Su, C., Zhang, T. Photothermal-assisted triphase photocatalysis over a multifunctional bilayer paper. *Angew. Chem.* **133**, 23145-23151 (2021).
2. Yu, X., Bi, J., Yang, G., Tao, H., Yang, S. Synergistic effect induced high photothermal performance of Au nanorod@Cu₇S₄ yolk-shell Nanooctahedron particles. *J. Phys. Chem. C* **120**, 24533-24541 (2016).
3. Sun, M., Fu, X., Chen, K., Wang, H. Dual-plasmonic gold@copper sulfide core-shell nanoparticles: Phase-selective synthesis and multimodal photothermal and photocatalytic behaviors. *ACS Appl. Mater. Interfaces* **12**,

46146-46161 (2020).

4. Li, Y. *et al.* Coupling resonances of surface plasmon in gold nanorod/copper chalcogenide core–shell nanostructures and their enhanced photothermal effect. *ChemPhysChem* **19**, 1852-1858 (2018).
5. Tao, F. *et al.* From CdS to Cu₇S₄ nanorods via a cation exchange route and their applications: Environmental pollution removal, photothermal conversion and light-induced water evaporation. *ChemistrySelect* **2**, 3039-3048 (2017).
6. Cao, Y. *et al.* Rattle-type Au@Cu_{2-x}S hollow mesoporous nanocrystals with enhanced photothermal efficiency for intracellular oncogenic microRNA detection and chemo-photothermal therapy. *Biomater.* **158**, 23-33 (2018).
7. Ma, S. *et al.* Controlled growth of CdS–Cu_{2-x}S lateral heteroshells on Au nanoparticles with improved photocatalytic activity and photothermal efficiency. *J. Mater. Chem. A* **7**, 3408-3414 (2019).
8. Shanmugam, V. *et al.* Oligonucleotides—assembled Au nanorod-assisted cancer photothermal ablation and combination chemotherapy with targeted dual-drug delivery of doxorubicin and cisplatin prodrug. *ACS Appl. Mater. Interfaces* **6**, 4382-4393 (2014).
9. Hans, E.A.D.R., Regulacio, M. D. Dual plasmonic Au–Cu_{2-x}S nanocomposites: Design strategies and photothermal properties. *Chem. Eur. J.* **27**, 11030-11040 (2021).
10. Ivanchenko, M., Jing, H. Smart design of noble metal–copper chalcogenide dual plasmonic heteronanoarchitectures for emerging applications: Progress and prospects. *Chem. Mater.* **35**, 4598-4620 (2023).
11. Ma, L. *et al.* Pt decorated (Au Nanosphere)/(CuSe ultrathin nanoplate) tangential hybrids for efficient photocatalytic hydrogen generation via dual-plasmon-induced strong Vis–NIR light absorption and interfacial electric field coupling. *Sol. RRL* **4**, 1900376 (2020).
12. Sen, S., Shyamal, S., Mehetor, S.K., Sahu, P., Pradhan, N. Au-Cu_{2-x}Te Plasmonic heteronanostructure photoelectrocatalysts. *J. Phys. Chem. Lett.* **12**,

11585-11590 (2021).

13. Ivanchenko, M., Nooshnab, V., Myers, A.F., Large, N., Evangelista, A.J., Jing, H. Enhanced dual plasmonic photocatalysis through plasmonic coupling in eccentric noble metal-nonstoichiometric copper chalcogenide hetero-nanostructures. *Nano Res.* **15**, 1579-1586 (2022).
14. Chen, K. *et al.* Tunable charge transfer and dual plasmon resonances of Au@WO_{3-x} hybrids and applications in photocatalytic hydrogen generation. *Plasmonics* **15**, 21-29 (2020).
15. Ma, L. *et al.* Growth behavior of Au/Cu_{2-x}S hybrids and their plasmon-enhanced dual-functional catalytic activity. *CrystEngComm* **21**, 5610-5617 (2019).
16. Basu, M., Nazir, R., Fageria, P., Pande, S. Construction of CuS/Au heterostructure through a simple photoreduction route for enhanced electrochemical hydrogen evolution and photocatalysis. *Sci. Rep.* **6**, 34738 (2016).
17. Xu, Y. *et al.* Dual-plasmon-enhanced nitrophenol hydrogenation over W₁₈O₄₉-Au heterostructures studied at the single-particle level. *Catal. Sci. Technol.* **13**, 1301-1310 (2023).